# Annual variability of ice nucleating particle concentrations at different Arctic locations

Heike Wex[1], Lin Huang[2], Wendy Zhang[2], Hayley Hung[3], Rita Traversi[4], Silvia Becagli[4], Rebecca J. Sheesley[5], Claire E. Moffett[5], Tate E. Barrett[5], Rossana Bossi[6], Henrik Skov[6], Anja Hünerbein[1], Jasmin Lubitz[1], Mareike Löffler[1,a], Olivia Linke[1], Markus Hartmann[1], Paul Herenz[1,b], and Frank Stratmann[1]

[1]Experimental Aerosol and Cloud Microphysics, Leibniz Institute for Tropospheric Research (TROPOS), Leipzig, Germany
[2]Climate Research Division, Atmospheric Science & Technology Directorate, STB, Environment & Climate Change Canada, Toronto, Canada
[3]Air Quality Processes Research Section, Environment & Climate Change Canada, Toronto, Canada
[4]Department of Chemistry "Ugo Schiff", University of Florence, Florence, Italy
[5]Department of Environmental Science, Baylor University, Waco, Texas, US
[6]Department of Environmental Science, iCLIMATE, Aarhus University, Roskilde, Denmark
[a]now at: Deutscher Wetterdienst, Zentrum für Agrarmeteorologische Forschung Braunschweig (ZAMF), Braunschweig
[b]now at: Senate Department for the Environment, Transport and Climate Protection, Berlin, Germany.

**Correspondence:** Heike Wex (wex@tropos.de)

**Abstract.** Number concentrations of ice nucleating particles ($N_{\mathrm{INP}}$) in the Arctic were derived from ground-based filter samples. Examined samples had been collected in Alert (Nunavut, Northern Canadian Archipelago on Ellesmere Island), Utqiaġvik, formerly known as Barrow (Alaska), Ny Ålesund (Svalbard) and at the Villum Research Station (VRS, Northern Greenland). For the former two stations, examined filters span a full yearly cycle. For VRS, 10 weekly samples, mostly from different months of one year, were included. Samples from Ny Ålesund were collected during the months from March until September of one year. At all four stations, highest concentrations were found in the summer months from roughly June to September. For those stations with sufficient data coverage, an annual cycle can be seen. The spectra of $N_{\mathrm{INP}}$ observed at the highest temperatures, i.e., those obtained for summer months, showed the presence of INP that nucleate ice up to -5°C. It is known fromAlthough the nature of these highly ice active INP could not be determined in this study, it often has been described in literature that ice activity observed at such high temperatures indicates originates from the presence of ice active material of biogenic origin. Spectra observed at the lowest temperatures, i.e., those derived for winter months, were on the lower end of respective values reported infrom literature on Arctic INP and INP from mid-latitude continental sites, to which a comparison is presented herein. An analysis concerning the origin of INP that were ice active at high temperatures was carried out, using back-trajectories and satellite information. Both, terrestrial locations in the Arctic and the adjacent sea were found to be possible source areas for highly active INP.

# 1 Introduction

The Arctic warms faster than any other region on Earth, a phenomenon which is known as Arctic amplification (Serreze and Barry, 2011; Cohen et al., 2014; IPCC, 2013). Many different processes, of which some are heavily interconnected, contribute to this (Pithan and Mauritsen, 2014). However, not all of these processes and feedbacks are fully understood, and some might even still be unknown. Clouds in the Arctic are special in that they often form extended, persistent, low-level stratiform cloud layers, which are kept stable for days by different feedback processes (Shupe et al., 2006; Morrison et al., 2012; Shupe et al., 2013). These clouds influence the energy budget and generally warm the surface, compared to clear skies (Intrieri et al., 2002). They often contain supercooled liquid water. In the range of temperatures ($T$) down to -20°C, fractions of supercooled liquid clouds were reported to be well above 50%, based on annual mean data for Europe and North America (both including the Arctic) from satellite remote sensing (Choi et al., 2010). For a multi-year analysis of all clouds, based on ground-based remote sensing at two western Arctic locations (Eureka and Utqiaġvik), clouds containing only liquid water occurred at least 20% of the time in all months with a maximum of 56% in September (Shupe, 2011). Also during two Arctic aircraft campaigns operating out of Inuvik, each in April and May of two different years, based on in-situ measurements, at least 60% of the clouds observed down to -18°C were characterized as mostly liquid (Costa et al., 2017). ground-based and satellite remote sensing (Shupe, 2011; Choi et al., 2010) as well as on aircraft measurements (Costa et al., 2017).

For primary ice formation in clouds, ice nucleation has to occurIce nucleation forms primary ice in clouds, and for $T$ from 0°C to roughly -38°C, ice nucleating particles (INP) are needed to induce this nucleation process. Not many measurements on number concentrations of INP ($N_{\mathrm{INP}}$) in the Arctic existwere done up to now, however, these particles play an important role in the lifetime and radiative effects of Arctic stratiform clouds. A number of effects of ice in clouds are described in Prenni et al. (2007), among them that ice clouds are optically thinner than supercooled liquid water clouds so that the former emit less longwave radiation towards the surface. A modeling study showed that an increase in $N_{\mathrm{INP}}$ may cause a faster dissipation of these stratiform clouds, (Loewe et al., 2017), which in turn will influence the surface energy budget. But it should also be mentioned that $N_{\mathrm{INP}}$ of $> 1\,\mathrm{L}^{-1}$ were needed to obtain an effect in this modeling study, and such concentrations were observed for mid-latitude regions only for $T$ below $\approx$ -15°C (Petters and Wright, 2015; O'Sullivan et al., 2018). Recycling of INP was assumed to be possible in Arctic clouds, again based on a modeling study (Solomon et al., 2015), i.e., ice crystals falling from a cloud could sublimate and re-entrain into clouds from below, which might potentially enhance the effect of changes in $N_{\mathrm{INP}}$. It also has been shown with large-eddy simulations that ice crystal number concentrations significantly influence cloud structure and evolution of Arctic mixed-phase clouds (Ovchinnikov et al., 2014). Overall, the cloud phase (i.e., supercooled water versus ice) is important for the radiation budget and hence the effect of Arctic stratiform clouds on climate. Kalesse et al. (2016) examined in detail a mixed-phase stratiform Arctic cloud and its phase transitions for roughly 1.5 days. They used a large set of measurements and applied modelling to gain further understanding. Observed changes in the cloud were related to changes in air mass, but it was also said explicitly that for a better understanding of cloud phase transitions, among other observations also measurements of $N_{\mathrm{INP}}$ are needed. All of this highlights the importance of insight on the abundance of INP in the Arctic and on their sources and sinks.

In the past, some studies on Arctic INP were done. However, most studies only included samples collected during short term deployments. A comparison of results of the present study with some of those from literature will be made further down in Sec. 4, while the main outcomes of these previous studies are already described in the following. In general, it can still be said that data on Arctic $N_{\mathrm{INP}}$ is scarce, which is particularly true for data at high $T$. Borys (1983) and Borys (1989) derived $N_{\mathrm{INP}}$ based on ground-based and aircraft measurements, respectively. It was suggested that mid-latitude pollution did not contribute INP to the Arctic aerosol, as $N_{\mathrm{INP}}$ were found to be lowest in winter, when Arctic haze, originating from anthropogenic pollution, was present. Bigg (1996) measured INP during a ship cruise in the Arctic in the months from August to October. It was concluded that INP were oceanic in origin while land was only a weak source, and that the upper troposphere was deficient in INP. Bigg and Leck (2001) derived $N_{\mathrm{INP}}$ also during an Arctic ship cruise in July to September, and found a decline in $N_{\mathrm{INP}}$ during that phase. At least for some of the detected INP, the most likely sources were assumed to be bacteria and fragments from marine biota emitted via bubble bursting from the open sea. Rogers et al. (2001) detected INP during aircraft measurements in the Arctic during the month of May. They reported strongly varying concentrations and found some INP that contained Si that were likely mineral dust particles, while other INP seemed to consist of low-molecular-weight components. In Prenni et al. (2007), also aircraft measurements of $N_{\mathrm{INP}}$ were made in the vicinity of Arctic clouds during fall. They found, reporting lower values than Rogers et al. (2001) had obtained in spring. and theyIt was concluded that typical Arctic values for $N_{\mathrm{INP}}$ might be overestimated by current parameterizations. Mason et al. (2016) derived $N_{\mathrm{INP}}$ for size segregated aerosol samples collected between end of March until July in Alert. $N_{\mathrm{INP}}$ derived from these Alert samples were slightly below values reported for other more southerly stations (mostly in Canada) in that study. They also found that at all stations large fractions of INP were contributed by supermicron particles. These fractions generally were largest at the highest $T$ at which measurements were made, i.e., at -15°C, where, for the Alert samples, > 90% and 70% of all INP were > 1 $\mu$m and > 2.5 $\mu$m, respectively. Similarly, Si et al. (2018) found a size dependent ability of particles nucleating ice for samples collected mostly in coastal areas in southern Canada and one sample collected in Lancaster Sound in the Canadian Arctic, with larger particles being more ice active. They also concluded that sea spray aerosol was not a major contributor to INP for the samples taken in southern Canada. Based on concentrations of K-feldspar taken from a global model, $N_{\mathrm{INP}}$ measured at -25°C were modelled well, while INP ice active at -15°C were missing in this model. Also Creamean et al. (2018) foundreport a strong size dependence of the ice activity in samples collected on land in the northern Alaskan Arctic, where again the largest particles in the supermicron size range were the most efficient INP. During sampling phases from March until mid May, when grounds were covered in snow and ice, number concentrations of the supermicron INP were lower than those in late May, by up to two orders of magnitude. The increase in $N_{\mathrm{INP}}$ was suggested to originate from open leads in the sea ice and from open tundra. Similarly, for a coastal mountain station in northern Norway (at 70°N), based on four filters sampled during July, Conen et al. (2016) observed that air masses were enriched in INP ice active at -15°C when they had passed over land. An origin of these INP from decaying leaves was suggested. Irish et al. (2019) derived $N_{\mathrm{INP}}$ during a ship cruise in the Canadian Arctic marine boundary layer in summer. They suggest that mineral dust contributed more strongly to the observed INP than sea spray, with mineral dust particles likely originating in the Arctic (Hudson Bay, eastern Greenland, north-west continental Canada).

Different substances are known to contribute to atmospheric INP, as given in a number of review articles (Szyrmer and Zawadzki, 1997; Hoose and Moehler, 2012; Murray et al., 2012; Kanji et al., 2017). In general, it is known that $N_{INP}$ increases roughly exponentially with decreasing $T$, although at higher $T$ steep increases may be observed, followed by a weaker increase or even a plateau region down to roughly -20°C (as seen in e.g., Petters and Wright, 2015; O'Sullivan et al., 2018; Creamean

et al., 2019). Ice nucleation at higher $T$ typically is related to macromolecules from biogenic entities as bacteria, fungal spores, lichen, pollen and marine biota. These ice active macromolecules nucleate ice from just below 0°C down to roughly -20°C (Murray et al., 2012; Kanji et al., 2017; O'Sullivan et al., 2018). But Biogenic INP typically occur in low concentrations in the atmosphere, but nevertheless, at remote marine locations as the Southern Ocean, where $N_{INP}$ is generally low, marine biogenic INP might make up a large fraction or even the entire INP population (Burrows et al., 2013; McCluskey et al., 2018a).

At less remote locations, the majority of atmospheric INP consists of mineral dust particles originating from deserts or soils. Pure mineral dust particles of atmospheric relevant sizes typically are ice active below -15°C (Murray et al., 2012; Kanji et al., 2017) or even below -20°C (Augustin-Bauditz et al., 2014) and, with the above mentioned exception of remote marine locations, typically occur at much higher concentrations than biogenic INP (Murray et al., 2012; Petters and Wright, 2015). However, mineral dust particles might also occur together with biogenic ice active material (Tobo et al., 2014; O'Sullivan

et al., 2014; Hill et al., 2016), and such a mixed particle acts like a biogenic INP (Augustin-Bauditz et al., 2016) and should be attributed to the aforementioned group of biogenic INP.

The existence of particularly high fractions of supercooled water observed in Arctic stratiform clouds, as e.g., observed in Costa et al. (2017) in the temperature range above -20°C, in comparison to more convective clouds in midlatitudes and the tropics, could be expected to be linked to a lack of biogenic INP in the Arctic, due to sparse biological activity. However,

it is known that biogenic INP are contained in sea water (Schnell, 1977) and the oceanic surface microlayer (SML) (Wilson et al., 2015; Irish et al., 2017) and are emitted to the atmosphere by sea spray production (DeMott et al., 2016). An increase in INP concentrations in the SML (Wilson et al., 2015) and for the biosphere in general (Schnell and Vali, 1976) from equatorial regions towards the poles has been observed. Also, present in the Arctic are fungi (Fu et al., 2013), lichen and bacteria (Santl-Temkiv et al., 2018) which could potentially contribute biogenic INP.

In the present study, we aimed at increasing the knowledge on Arctic surface concentrations of INP active in the immersion freezing mode, as described in the following. The immersion freezing mode was examined as it has been described as the most important heterogenous ice nucleation mode in mixed-phase clodus (Ansmann et al., 2009; Wiacek et al., 2010; de Boer et al., 2011). No specific measurement campaign was organized to do the examinations described herein. Instead, use was made of already existing filter samples. Besides for deriving temperature spectra of $N_{INP}$ for 104 filter samples (Sec. 3.1), we also

determined possible sources for INP that are ice active at high $T$ for selected samples. For that, correlations between $N_{INP}$ and some chemical compounds were made (Sec. 3.2.1), and an analysis was done concerning possible regions of origin of INP that are ice active at high $T$ (Sec. 3.2.2) on a selection of samples. The results will also be discussed in light of literature data (Sec. 4).

## 2  MethodsMeasurements

Quartz fiber filters were sampled regularly at the four Arctic measurement stations of Alert, Ny Ålesund, Utqiaġvik, and Villum Research Station (VRS) during the past years. Fig. 1 shows the locations of these four stations, which are all in close proximity to the ocean (<3 km). A portion of the filters was provided for the herein presented analysis. In the following, some detail will be given on these four different stations, including the filter handling, and on the measurement and evaluation method used to obtain INP number concentrations. We also describe in detail the temperature history of the filters, although it is not known with certainty up to date how the temperature during storage will affect INP concentrations. Generally, the filters were kept frozen whenever possible. Transport from the four institutes where the samples had been kept to TROPOS was done in insulated boxes, together with cooling elements. The shipment was organized such that transport was fast (one to three days) and that upon arrival at TROPOS the temperature in the boxes was still below 0°C. At TROPOS, samples were again stored at -18°C until the measurements were done. These measures during storage and transport are precautions, as for biogenic INP, storage at temperatures above 0°C or even storage under freezing conditions has been described to reduce their ice activity (Wex et al., 2015; Polen et al., 2016, respectively). In this study, until mentioned otherwise, all samples from all stations (including also field blanks) were treated similar during all procedures. In the next sections, peculiarities of the separate four stations are described, followed by details on the measurements and its evaluation.

### 2.1  Alert

A custom-built high-volume aerosol sampler was used at the Dr. Neil Trivett Global Atmosphere Watch Observatory in Alert, Canada (82°30′N 62°22′W, 210 m above sea level (a.s.l.)), to collect 38 samples between April 2015 and April 2016. The sampler is installed at a walk-up deck, about 4 m above the ground. Flow rate is approximately 1.4 m³/min at STP condition. Quartz filters (Pall Life Sciences, PallFlex Filters, 8 x 10 in, USA) were pre-fired at 900°C overnight and then shipped to Alert while already being loaded on cartridges. During transport and storage, the filter containing cartridges were wrapped by aluminum foil and they were inside of sealed plastic bags. Sampling time on those filters was either one week or two weeks, the latter being used from August until October (due to operational issues, no filter was sampled in July). A total of 9 field blanks (roughly one every month) were collected. These field blanks were treated similar to the other filters, i.e., inserted into the sampler, however, without an airflow through them. They were also stored similarly to the sampled filters at all times. After sampling, filters were stored (at room temperature ≈ 20°C) in their sampling cartridges (wrapped in aluminum foil inside sealed plastic bags) at the Alert station and shipped in cardboard boxes (containing 5 sampling cartridges each) to the Toronto lab at Environment Climate Change Canada where they were stored frozen at < -30°C. From these filters, a circular piece with 47 mm in diameter was providedshipped to Leipzig for this study.

The total sampling area on the filters was 17.8 cm x 22.8 cm. For the measurements at TROPOS, described in detail in Sec. 2.5 below, circles with 1 mm diameter were punched out from the samples using sterile biopsy punches, and immersed in ultra-pure water separately. The volume of air sampled per 1 mm piece of filter wasdiffered for the different samples and varied from roughly 270 to 540 L. transport to TROPOS was done in a cooler, together with cooling elements, where preparations

were done such that transport took less than 3 days and the box was stored at freezing temperatures during delay times at airports and customs. Upon arrival at TROPOS the temperature was just around 0°C. At TROPOS, samples were again stored at -18°C until the analysis was done.

## 2.2 Ny Ålesund

Filter sampling in Ny Ålesund on Svalbard (at 78°55′N 11°55′E, 10 m a.s.l.) is done by the University of Florence, Italy. Quartz fiber filters have been sampled regularly since 2010, using a high-volume sampler with quartz microfiber filters (CHMLAB GROUP QF1 grade, Barcelona, Spain). The filters were pre-treated at 400°C prior to sampling. The filters had a diameter of 47 mm, of which one quarter was provided for the present study. Sampling duration was 4 days in 2012, and of these filters, 13 sampled from late March until beginning of September were examined in the present study, together with two field blanks. The total air volume collected on each filter was roughly 200 m$^3$. Each circular 1 mm filter piece used for the analysis sampled particles from roughly 130 L. Once sampled, filters were stored in a freezer at the Italian base in Ny Ålesund and then shipped to Italy via cargo. At the home university, they were then stored in a cold room at -20°C. Transport to TROPOS was done in a Styrofoam box, together with cooling elements. The transport took little more than 24 hours and temperature inside the box was well below 0°C upon arrival. At TROPOS, samples were again stored at -18°C until the analysis was done.

## 2.3 Utqiaġvik (formerly known as Barrow)

Filter sampling in Utqiaġvik, Alaska (at 71°18′N, 156°46′W, 11 m a.s.l.) is done by Baylor University, US as described in Barrett and Sheesley (2017). For the present study, quartz fiber filters were used that had been sampled regularly in an annual campaign from June 2012 to June 2013, using a high-volume sampler (Tisch Environmental, Cleves, OH, USA). Filters were stored frozen prior to and immediately following all sampling. Two rectangular filter pieces (with a lateral length of 1.5 cm) from each of 41 different filters and two field blanks were provided for the present study. The sampled area of each filter was 399 cm$^2$. Sampling on each filter was done for 4 up to 13 days (7 days on average), collecting particles from a total air volume of roughly 6000 to 23000 m$^3$. This yields an air volume of roughly 120 to 450 L collected on each circular 1 mm filter piece used for the analysis. Prior to sampling, filters were pre-treated at 500°C. After sampling, the filters were stored in a freezer on site and transported to the home university in coolers with cooling elements, where they were then stored at -18°C. Transport to TROPOS was done in a cooler, together with cooling elements, where preparations were done such that transport took less than 2 days and the box was stored at freezing temperatures during delay times at airports and customs, such that filters arrived still well frozen. At TROPOS, samples were again stored at -18°C until the analysis was done.

## 2.4 Villum Research Station

Villum Research Station (VRS) at Station Nord in northern Greenland (at 81°36′N, 16°40′W, 10 m a.s.l.) is operated by Aarhus University, Denmark (in cooperation with the Danish Defense, the Arctic Command). Quartz fiber filters are sampled regularly since 2008, using a high-volume sampler (Digitel, Hegnau, Switzerland), employing weekly sampling Bossi et al. (2016). The

filters had an exposed area of 154 cm$^2$ and sampled a total air volume of roughly 5000 m$^3$. From filters sampled in 2015, a 2 cm diameter piece was cut from each of the filters and provided for this study. Due to the large interest in shares of the filters, only samples from 11 different filters, all from different months in 2015 and one from December 2013, could be used herein. As for all samples used in this study, from the 2 cm pieces punches of 1 mm in diameter were cut at TROPOS directly

prior to doing the measurements. The resulting small pieces were then used for INP analysis. The area of these 1 mm pieces corresponds to a sample volume of 255 L of air. Prior to sampling in the field, filters were pre-treated at 450°C. Storage of the filters at VRS was done in freezers. Filters are transported from Greenland to Denmark around 3 times per year by the Danish Royal Air Force and then shipped to Roskilde, where they were then stored at -18°C. Transport to TROPOS was done in a Styrofoam box, together with cooling elements, where the transport took roughly one day, and the temperature inside the box

was well below 0°C upon arrival. At TROPOS, samples were again stored at -18°C until analysis was done.

## 2.5   Freezing device INDA, the Ice Nucleation Droplet Array

For freezing experiments examining immersion freezing, a device comparable to one introduced in Conen et al. (2012) was used, but deploying PCR-trays (Hill et al., 2016) instead of separate tubes. The same device had been used in Chen et al. (2018). From each filter piece that had been shipped to TROPOS, circles with a diameter of 1 mm were punched out directly

before measurements were done, and each of the 96 wells of a PCR tray was filled with such a filter piece together with 50 $\mu$L of ultra-pure water. (Background measurements of ultra-pure water are given in the supporting information (SI)). After sealing the PCR tray with a transparent foil, it was immersed into a bath thermostat such that the water table in the wells was below the surface of the liquid in the thermostat. The bath of the thermostat was then cooled with a cooling rate of 1 K/min, and the freezing process was monitored by a camera, taking a picture every 6 seconds. A LED light source installed below the PCR

tray ensured that wells in which the water was still liquid could be easily distinguished from frozen ones. Frozen fractions ($f_{\text{ice}}$) were then determined as the number of frozen tubes divided by the total number of tubes. Typically, fresh water to be used in the experiments was taken once a day and stored in a glass bottle. Whenever fresh water was taken, an experiment was run with this water in the tubes only, to assure that the water was satisfyingly clean. Similarly, experiments were run with field blank filters that had the same history as the samples but without sampling (see SI), and signals from the field blanks were well

below those of the sampled filters. A subtraction of the signals of the field blank from those of the measurements was not done. This is justified in a detailed discussed in the SI. The interpretation of the results from the filters presented in this study is the same for both uncorrected and background-corrected samples.

## 2.6   Deriving $N_{\text{INP}}$

Equation 1 was used to derive $N_{\text{INP}}$ from the measured $f_{\text{ice}}$ (Vali, 1971; Conen et al., 2012). This equation accounts for

the possibility of the presence of multiple INP in one vial by assuming that the INP are Poisson distributed. Additionally it normalizes the values resulting from the measurement with the air volume sampled on each 1 mm filter piece. This yields

concentrations of ice nucleating particles per volume of sampled air:

$$N_{\mathrm{INP}} = -(ln(1 - f_{\mathrm{ice}}))/(F * A_{\mathrm{p}}/A_{\mathrm{f}}) \tag{1}$$

$F$ is the total volume of air drawn through the filter, $A_{\mathrm{p}}$ and $A_{\mathrm{f}}$ are the surface area of a single 1 mm filter piece and the whole sampled area of the filter, respectively.

The temperature and concentration regions for which data were obtained for the different samples depend on a number of factors. The measured value $f_{\mathrm{ice}}$ is a fraction ranging from 0 to 1. Therefore, $N_{\mathrm{INP}}$, as derived using equation 1, can only take on a limited range of values. This range is based on the negative natural logarithm of 0.01 and 0.99 (4.6 to 0.01). The absolute values of $N_{\mathrm{INP}}$ then also depend on the volume of air sampled onto each 1 mm filter piece, i.e., on the volume of air drawn through the filter during the sampling period and on the relation of the surface area of one filter piece to the total sampled

surface area. The volume collected per 1 mm filter piece was within a factor of 4.5 for all filters (120 to 540 L). Altogether, the range of $N_{\mathrm{INP}}$ that can be obtained herein is roughly from $2 \cdot 10^{-5}$ L$^{-1}$ to 0.04 L$^{-1}$. From this limitation, it also follows that also the range of $T$ for which $N_{\mathrm{INP}}$ could be obtained is limited, as it is tied to the concentrations that can be measured. Times with more ice active INP show up as $N_{\mathrm{INP}}$ at higher $T$. The highest $T$ at which ice activity was observed was close to -5°C, as will be shown in the next section.

It should also be noted that samples that had less than 60 L of air volume collected on each 1 mm filter piece were also examined, but $f_{\mathrm{ice}}$ was close to the background and therefore these samples were not considered in this study. Results from background measurements are given in the SI. Measurement uncertainty as shown in this work was derived based on Harrison et al. (2016), i.e., following the assumption that the INP are Poisson distributed between the different examined droplets. The first few droplets that freeze in each experiment therefore show the highest uncertainties.

## 2.7   Using back-trajectories and satellite maps

A more in depth analysis concerning possible INP source regions was done for a selection of filter samples from each measurement station. For that, 5-day back-trajectories were calculated with HYSPLIT (Stein et al., 2015) to determine the origin of sampled air masses. These calculations were based on GDAS (Global Data Assimilation System) meteorological data, using an hourly time resolution. A new trajectory was started every 6 hours during the whole time for which sampling was done on the

respective filter. Back-trajectories were initiated at an altitude of 100 m above the sampling locations, as this altitude still has a high likelihood of being connected to the ground, and as lower elevations are more prone to uncertainties. In the supplementary information (SI), these back-trajectories are shown separately for the selected examined filter samples.

Using these back-trajectories, it was examined over which ground the air masses collected on the filters had passed in the 5 days prior to arrival at the measurement station. The aim was to see contributions from open land or open water for the different

selected filter samples. Therefore, it was distinguished between snow, open land, sea ice and open water. To do so, maps from the Interactive Multisensor Snow and Ice Mapping System (IMS) (Helfrich et al., 2007; National-Ice-Center, 2008) were used. IMS maps are a composite product produced by NOAA/NEDIS (National Oceanic and Atmospheric Administration's National Environmental Satellite Data and Information Service) combining information on both sea ice and snow cover. Information

from 15 different sources of input are included in the production of these maps (Helfrich et al., 2007). Currently these maps have been provided for 20 years. We used the daily northern hemisphere maps with a resolution of 4 km (National-Ice-Center, 2008). For each time step we applied nearest-neighbor interpolation in space and time to find the corresponding satellite coordinate along the back-trajectory. With that, for each back-trajectory, we determined the conditions on ground during the passage of the air mass, i.e., if the ground was covered by snow or ice or if open water or open land was present. The resulting information is shown exemplarily in Fig. 4 for the filter sample collected in Alert starting June 10, 2015. Using these maps, it was counted how often air masses that were collected on one filter were above open land or open water while the air mass was below 100 m. (As will be discussed in detail in Sec. 3.2.2, for Utqiaġvik, results reported here always refer to an upper altitude of 500 m and 10-day back-trajectories.) An altitude restriction was used as we were trying to geographically locate INP sources on the surface. Additionally, the back-trajectories were only considered back in time until an integral amount of 2 mm of precipitation (taken from the information included in the back-trajectories) was reached. This was done as precipitation formation occurs via the ice phase, so that precipitation is assumed to lead to a wash out of INP.

## 3 Results

In the following, $N_{\mathrm{INP}}$ derived from filter samples will shortly be introduced. A correlation to some available chemical composition data is made. Finally an analysis on air mass origins is introduced to analyze possible source regions for INP that are ice active at high $T$. A comparison to literature data and further discussion of the results are then presented in Sec. 4 and 5, respectively.

### 3.1 Arctic atmospheric INP concentrations

Quartz fiber filters from the four different Arctic stations shown in Fig. 1 were analyzed to derive $N_{\mathrm{INP}}$. Fig. 2 shows $N_{\mathrm{INP}}$ for all different samples, separately for the four sampling locations. Due to the comparably large number of filters analyzed for Alert and Utqiaġvik, separate curves can not be seen easily in Fig. 2. Therefore, Fig. 3 shows time-series of $N_{\mathrm{INP}}$ for $T$ at -7°C, -10°C, -13°C and -9.5°C for Alert, Utqiaġvik, Ny Ålesund and VRS, respectively. $T$ was chosen such that $N_{\mathrm{INP}}$ could be obtained for the largest possible number of all curves (data for the three samples with the lowest $N_{\mathrm{INP}}$ are missing for Alert (April 29 and May 27, 2015 and April 4, 2016), and the one with the highest $N_{\mathrm{INP}}$ for Utqiaġvik (May 3, 2013), indicated by arrows in Fig. 3). The yellow background shows for which samples an more in-depth analysis is presented in Sec. 3.2.2. Error bars in Fig. 3 show the 95% confidence interval.

It is worth noticing that once a sample has a comparably high concentration at one $T$ this will be generally trueis generally observed at all $T$ at which measurements are available, and vice versa, i.e., curves do not intersect much (see Fig. 2). Therefore, the curves shown in Fig. 3 can be used to discuss observed trends for INP that are ice active at high $T$. It should also be noted that for Ny Ålesund, data only exists for March until September, and that for VRS, there is mostly only one data-point per month, if any (no data exists for February, March and May).

In Figs. 2 and 3 it can be seen that in general $N_{\mathrm{INP}}$ obtained for the summer months is higher than for the winter months. A decrease of $N_{\mathrm{INP}}$ is observed starting in fall (October or November). For months with the lowest observed ice activity, which generally are winter and early spring months, values for $N_{\mathrm{INP}}$ were measured down to below -20°C. From June until September mostly INP that were ice active between -5°C and -15°C were detected (see Fig. 2), and for Utqiaġvik and VRS such highly ice active INP were observed as early as in April. These highly ice active INP will be in the focus of this study in the next two sections (Sec. 3.2.1 and 3.2.2). Similarly Such highly ice active INP have been assumedsuggested to be biogenic in origin based on tests such as heat treatment (Hill et al., 2016; O'Sullivan et al., 2018), tests which due to the limited available amount of filter material could not be done in the present study. and these will be in the focus of this study in the next two sections (Sec. 3.2.1 and 3.2.2).

## 3.2 Sources of INP

### 3.2.1 Correlation to chemical composition

Typical $N_{\mathrm{INP}}$ measured in the atmosphere are several orders of magnitude below total particle number concentrations, and therefore mass concentrations of INP are so small that a correlation between bulk chemical composition and $N_{\mathrm{INP}}$ might not be expected, particularly not for the very rare INP that are ice active at high $T$. This is in line with recent findings for a long term study on INP at Cape Verde by Welti et al. (2018). There, no correlation between $N_{\mathrm{INP}}$ and bulk chemical composition was found for $T$ down to -16°C for a number of different compounds, which included $Ca^{2+}$, $Na^+$ and elemental carbon as tracers of continental, marine and combustion sources, respectively. At a lower $T$ of -25°C, recently Si et al. (2019) reported that mineral dust tracers correlated with INP, which suggests that mineral dust was a major contributor to the INP population at that $T$. Si et al. (2018) found that for three coastal sites in Canada a model based on K-feldspar as the only INP calculated $N_{\mathrm{INP}}$ that fit measurements well at -25°C while at -15°C measurements were under-predicted, suggesting a missing source of INP that are active at higher $T$. In the following it will be examined if there is a correlation between the INP detected in the present study that are ice active at high $T$ and chemical composition.

The examined filters pieces were not particularly sampled for this study, and were entirely needed to do the above described INP analysis. No dedicated chemical analysis could be done additionally. But as other parts of most of the filters were also used in other studies, some information on chemical composition was available. This was used to derive correlations with $N_{\mathrm{INP}}$ shown in Fig. 3, i.e., with those INP that are ice active at high $T$. Tab. 1 shows values for R, $R^2$ and p for linear correlations between $N_{\mathrm{INP}}$ and different bulk chemical properties.

In general, no correlations were found. The highest values for $R^2$ ($> 0.3$) together with values for p $< 0.05$ were found for potassium and sulphate in Ny Ålesund and The only case with a positive value for R (0.59) and a low value for p (0.01) was found for POC+CC (pyrolyzed organic carbon and carbonate carbon) in Alert. While it was a positive correlation between $N_{\mathrm{INP}}$ and POC+CC, there were anti-correlations between $N_{\mathrm{INP}}$ and both potassium and sulphate. And even these highest values for $R^2$ are still low. Nevertheless, a short explanation should be given on POC+CC. Both POC and CC contain carbon that pyrolyzes at 870°C in a pure He stream (Huang et al., 2006). CC might indicate the presence of soil dust (Huang et al.,

2006). POC includes some charred carbon formed at 550°C, which is the lower temperature step of the applied analysis (EnCan-total-900 method, Huang et al., 2006; Chan et al., 2010) and highly oxidized organic compounds / high molecular weight refractory carbon. Based on previous studies, the POC mass is proportional to the oxygen mass in organic aerosols (Chan et al., 2010), releasing as carbon monoxide at 870°C. POC was observed to form from sucrose and glucose (Huang et al., 2006), and therefore might be indicative of biogenic material. This might points towards a direction in which more detailed studies cshould be undertaken in the future. In Sec. 5, we will discuss a range of possible sources for the observed INP.

$N_{\mathrm{INP}}$ are several orders of magnitude below total particle number concentrations, and therefore mass concentrations of INP are so small that a correlation between bulk chemical composition and $N_{\mathrm{INP}}$ might not be expected, particularly not for the very rare INP that are ice active at high $T$. This is in line with recent findings for a long term study on INP at Cape Verde by Welti et al. (2018). There, also no correlation between $N_{\mathrm{INP}}$ and bulk chemical composition was found for $T$ down to -16°C for a number of different compounds, which included $Ca^{2+}$, $Na^+$ and elemental carbon as tracers of continental, marine and combustion sources, respectively.

### 3.2.2 Determination of possible source regions

A more in depth analysis concerning possible INP source regions was done for a selection of filter samples from each measurement station. For that, samples were chosen that had been collected in spring, directly before and after the transition from typical winter to typical summer conditions (see yellow background in Fig. 3). 17 separate filter samples were included in the following analysis, two for VRS, four for Alert and Ny Ålesund each, and seven for Utqiaġvik.

5-day back-trajectories were calculated for all of these samples with HYSPLIT (Stein et al., 2015), based on GDAS (Global Data Assimilation System) meteorological data, using an hourly time resolution and starting a new trajectory every 6 hours during the whole time for which sampling was done on the selected filters. Back-trajectories were initiated at an altitude of 100 m above the sampling locations, as this altitude still has a high likelihood of being connected to the ground, and as lower elevations are more prone to uncertainties. In the supplementary information (SI), these back-trajectories are shown separately for the 17 examined filter samples.

Using these back-trajectories, it was examined over which ground the air masses collected on the filters had passed in the 5 days prior to arrival at the measurement station. The aim was to see if contributions from open land or open water were potentially more pronounced during times when INP active at high $T$ were observed. Therefore, it was distinguished between snow, open land, sea ice and open water. To do so, maps from the Interactive Multisensor Snow and Ice Mapping System (IMS) (Helfrich et al., 2007; National-Ice-Center, 2008) were used. IMS maps are a composite product produced by NOAA/NEDIS (National Oceanic and Atmospheric Administration's National Environmental Satellite Data and Information Service) combining information on both sea ice and snow cover. Information from 15 different sources of input are included in the production of these maps (Helfrich et al., 2007). Currently these maps have been provided for 20 years. We used the daily northern hemisphere maps with a resolution of 4 km (National-Ice-Center, 2008). For each time step we applied nearest-neighbor interpolation in space and time to find the corresponding satellite coordinate along the back-trajectory. With that, for each back-trajectory, we determined the conditions on ground during the passage of the air mass, i.e., if the ground was

covered by snow or ice or if open water or open land was present. The resulting information is shown exemplarily in Fig. 4 for the filter sample collected in Alert starting June 10, 2015. Using these maps, it was counted how often air masses that were collected on one filter were above open land or open water while the air mass was below 100 m. (As will be discussed in detail below, for Utqiaġvik, results reported here always refer to an upper altitude of 500 m and 10-day back-trajectories.) An altitude

restriction was used as we were trying to geographically locate INP sources on the surface. Additionally, the back-trajectories were only considered back in time until an integral amount of 2 mm of precipitation (taken from the information included in the back-trajectories) was reached. This was done as precipitation formation occurs via the ice phase, so that precipitation leads to a wash out of INP.

Results from the more in depth analysis concerning possible INP source regions, based on back-trajectories and satellite

maps as described in Sec. 2.7, is presented in the following. For this analysis, samples were chosen that had been collected in spring, directly before and after the transition from typical winter to typical summer conditions (see yellow background in Fig. 3). 17 separate filter samples were included, two for VRS, four for Alert and Ny Ålesund each, and seven for Utqiaġvik. The aim was to see if contributions from open land or open water were potentially more pronounced during times when INP active at high $T$ were observed.

Fig. 5 shows the number of time steps when air masses were over open land or open water for the separate filter samples. Additionally, gray bars in the background indicate the percentage of time the air masses collected on one filter were below 100 m for Alert, VRS and Ny Ålesund or below 500 m for Utqiaġvik. It can already be seen, that , besides for Utqiaġvik, the presence of highly ice active INP on a filter is related to air masses that fulfill the above given criteria, i.e., that traveled over open land or open water at a low altitude. It also can be seen that this was not found for Utqiaġvik. Initially, no open land and

hardly any open water had been found for this site when 5-day back-trajectories were used, together with an altitude restriction of 100 m, which means that air masses did not travel over open land or open water at altitudes below 100 m. To check if the length of the back-trajectory or the chosen maximum altitude influenced our results for Utqiaġvik, an analysis was also done using 10-day back-trajectories and 500 m , although here 10-day instead of 5-day back-trajectories were used, and although 500 m was taken as the altitude limit, presented in Fig. 5. This extension The latter simply only resulted in larger percentages

of time for which the air masses were below this altitude limit. But still not a large number of time steps was found for which air masses traveled over open land or open water for Utqiaġvik. We will get back to this again below.

Four separate rows in Fig. 6 (from A to D) show the spectra of $N_{\mathrm{INP}}$ (called INP spectra for simplicity from now on) for the samples included in this analysis (right panels) and the locations where the respective air masses traveled over open land or open water at altitudes below 100 m (or 500 m for Utqiaġvik) (left panels). Three different types of INP spectra can be

distinguished: First there are those for which we observed the start of ice activation only at around -10°C and which went down to well below -15°C. For these, INP spectra and locations are depicted in magenta or orange. Second, there are INP spectra for which we observed ice nucleation from roughly -5°C to above -15°C. For these, INP spectra and locations are depicted in greenish colors. The third category was used only for Utqiaġvik for INP spectra with medium ice activity depicted in blueish. Error bars shown in Fig. 6 show the 95% confidence interval.

For Alert, VRS and Ny Ålesund, the absence or scarcity of orange and magenta marks on the maps in Fig. 6 (maps on the left at A, B and C) shows that almost no open land or open water contributed to air masses sampled on the respective filters. This is in accordance with the corresponding INP spectra, which showed comparably low ice activity. The magenta locations close to Svalbard for the Alert sample correspond to the sample from May 20, 2015, for which somewhat more ice active INP were found than for the subsequent sample from May 27, 2015. For this latter sample, no contributions from open land or open water were observed, and it is, in fact, the sample with the lowest ice activity observed in this study (see Fig. 2). Locations depicted in greenish colors potentially contributed INP that are ice active at high $T$. They can be found on open land as well as on open water. In connection to filters sampled at Alert or at VRS they show up in North Greenland, on Ellesmere Island (on which Alert is located), in the Baffin Bay and along the southern part of the west coast of Greenland. Concerning filters sampled at Ny Ålesund, greenish marks show up on Svalbard and the adjacent sea.

The above analysis shows that coastal regions may be particularly important as source for highly ice active INP, including open waters close to coasts. Indeed, highly ice active biogenic INP were found in Arctic surface waters before (e.g., Wilson et al., 2015; Irish et al., 2017). For the highly ice active samples collected on Ny Ålesund on June 12 and June 28, 2012, the surrounding of the measurement station was completely snow free during the times when these samples were collected, whereas for all other cases there was at least partial or total snow cover around the stations. In other words, local terrestrial sources close to the measurement station may also contribute as sources for highly ice active INP, as already discussed in Creamean et al. (2018). Also Irish et al. (2019) describe Arctic land masses to be the source for observed Arctic INP (ice active at -15°C, -20°C and -25°C), and these INP were suggested to be mineral dust. On Svalbard, Tobo et al. (2019) found higher atmospheric $N_{\mathrm{INP}}$ in July than in March, and they additionally described glacial outwash sediments in Svalbard to be highly ice active. This ice activity was assumed to be connected to small amounts of organic (likely biogenic) material. Based on these findings, Tobo et al. (2019) suggest the higher $N_{\mathrm{INP}}$ in summer to be connected to organic (biogenic) components in glacially sourced dust. Some coastal regions in the Arctic, e.g., the west coast of Greenland together with the region around Baffin Bay and the Canadian Arctic Archipelago as well as the area around the Bering Strait and also Svalbard are known for their abundance of sea bird colonies (Croft et al., 2016). These regions partially coincide with regions highlighted as possible INP sources in Fig. 6. These regions are known to emit ammonia, which plays a role in new particle formation in the Arctic (Croft et al., 2016). But clearly, newly formed particles are not expected to contribute to atmospheric INP at the temperatures examined in this study, and INP are likely simply also emitted from regions with high biological activity. In Sec. 5 we will discuss possible INP sources in more detail.

Some regions in the Arctic are known for their abundance of sea bird colonies, e.g., the west coast of Greenland together with the region around Baffin Bay and the Canadian Arctic Archipelago as well as the area around the Bering Strait and also Svalbard (Croft et al., 2016). These colonies emit ammonia which plays a role in new particle formation in the Arctic (Croft et al., 2016). Newly formed particles are not expected to contribute to atmospheric INP at the temperatures examined in this study, however, it is striking that these regions are also those that were found here as possible sources for highly ice active INP. The available open land does not only attract sea birds but also seems to enable the emission of highly ice active INP. Similarly, open water close to coasts did show up as possible locations contributing to highly ice active INP, too. Highly ice

For Utqiaġvik, altogether data from seven filters were included in the analysis. Two of them showed INP spectra at comparably low $T$, three at medium $T$ and two at high $T$. For all types of INP spectra, no contribution from open land was observed with the back-trajectory analysis. Only minor contributions from open water were found for the latter two types, although the

analysis was extended to include 10-day back-trajectories and the maximum altitude up to which air masses were considered was relaxed to 500 m. Air masses did travel below 100 m, and even more often below 500 m (see Fig. 5 and the back-trajectories for Utqiaġvik and their heights profiles in the SI). But the transition to filters on which INP active at comparably high $T$ were found to happen earlier at Utqiaġvik than at the other three measurement locations, already towards the end of March. IMS maps almost exclusively identified the ground as sea ice and snow in the regions that were crossed by the air masses, even until

beginning of May 2013. And in general, air masses spent more time over sea ice than over snow (see back-trajectories in the SI). There has to be a source for highly ice active INP that was not revealed in the analysis done here. Polynyas and open leads may contribute to explain this inconsistency. The resolution of the IMS maps used may be too coarse so that open water related to polynyas and open leads could have gone unnoticed.

While the analysis introduced here shows regions that potentially may have contributed highly ice active INP to the sampled

air masses, it does not make any statement about other regions. Other regions potentially could be sources, too, but might only have been crossed by air masses at high altitudes, or may not have been crossed at all.

## 4   Comparison with literature

Fig. 7 shows the ranges of $N_{\mathrm{INP}}$ observed for the four stations as shaded areas. Data from the different stations cover a rather similar range. As explained above (Sec. 2.6), $N_{\mathrm{INP}}$ could only be measured up to some $10^{-2}\,\mathrm{L}^{-1}$, depending on the volume

of air sampled onto one 1 mm filter piece. Hence the upper concentration limit of our data is determined by the measurement method. The gray background shows literature data of $N_{\mathrm{INP}}$ determined from precipitation samples collected mostly in North America and Europe (Petters and Wright, 2015). In that data set, those samples showing the highest $N_{\mathrm{INP}}$ at $T > $ -20°C originate from rain and hail samples collected in North Carolina (US) and Alberta, Montreal (CA). Some of the INP spectra we detected at the highest $T$, observed particularly in Alert and Utqiaġvik, show values for $N_{\mathrm{INP}}$ that are similar or only about

1 order of magnitude below data reported in Petters and Wright (2015). At the lowest temperatures at which we detected INP spectra, $N_{\mathrm{INP}}$ are lower than data from Petters and Wright (2015), i.e., $N_{\mathrm{INP}}$ in the Arctic may, at times, be lower than at other locations.lowest Arctic $N_{\mathrm{INP}}$, as those we observed in the winter months, might be below the lowest values observed on

continents in mid-latitudes. It is worthwhile adding that still lower concentrations were observed in marine remote locations in the Southern Ocean (McCluskey et al., 2018a) and for clean marine air in the North East Atlantic (McCluskey et al., 2018b).

Fig. 7 shows additional data on Arctic $N_{\text{INP}}$ from literature that was already discussed in the Introduction (Borys, 1983, 1989; Bigg, 1996; Bigg and Leck, 2001; Rogers et al., 2001; Prenni et al., 2007; Mason et al., 2016; Conen et al., 2016; DeMott et al., 2016). Not every single data point from these papers is shown, as Fig. 7 aims at giving an overview of the range of data that exists. Data on Arctic $N_{\text{INP}}$ in general is still scarce, which is particularly true for data at high $T$. Also, data scatters over a wide range. Highest values of $N_{\text{INP}}$ shown in Fig. 7 originate from one set of aircraft measurements made in May (Rogers et al., 2001), while a second set of aircraft measurements, taken in September and October on-board an aircraft flying out of Alaska, agrees with our data at the highest $T$ (Prenni et al., 2007). For the data taken from (Prenni et al., 2007), the reported measurement uncertainty is shown, representative for typical uncertainties for the type of instrumentation used in Rogers et al. (2001) and Prenni et al. (2007). Error bars indicate one standard deviation at the higher end. The lower end is indicative of the detection limit, and for a substantial fraction of measurements no INP were detected in Prenni et al. (2007).

Going back to Fig. 7, literature data from ground-based measurements that are in the same range of $N_{\text{INP}}$ in which we measured are also within the same $T$ range. But besides for data from Conen et al. (2016), these data are at the lower end of $T$ that we observed. This also holds for the data from DeMott et al. (2016), which were obtained during summer ship cruises in the Baffin Bay and in the central Bering Sea.

Data shown for Borys (1983) were taken in Ny Ålesund and Utqiaġvik. A slight tendency towards higher $N_{\text{INP}}$ in summer months, compared to winter months, can be seen. Similarly, as said in the Introduction, Bigg and Leck (2001) found a decreasing trend for $N_{\text{INP}}$ at -15°C from July to September. The observed decrease was roughly one order of magnitude, however, the scatter from sample to sample was almost as large as that trend. Nevertheless, these data sets are early indications of the annual trend that has been found very clearly in the present study.

Concerning possible sources for INP, Bigg (1996) assumed that mainly oceanic sources contributed to the observed INP, with only a weak contribution from land. Bigg and Leck (2001) discussed a marine origin of at least some of the INP they analyzed. These latter two studies were ship based. The land based study by Conen et al. (2016) showed INP that were ice active at higher $T$ than those observed in Bigg (1996) and Bigg and Leck (2001). Conen et al. (2016) traced these INP back to terrestrial contributions, possibly decaying leaves. During aircraft measurements, Rogers et al. (2001) identified some INP as mineral dust particles and others as containing low-molecular-weight components. These latter might have been connected to biogenic INP.

These different studies report very diverse $N_{\text{INP}}$. In these studies, different instrumentation was used and sometimes different ice nucleation mechanisms were probed. Also instrumental limitations typically determine the ranges of $T$ and $N_{\text{INP}}$ that can be probed. All of this might add to the diversity in the data. But, as can be seen in Fig. 3, a difference of up to 2 orders of magnitude in $N_{\text{INP}}$ was measured between winter and summer for a single temperature in our data set. Therefore, the diversity in $N_{\text{INP}}$ reported in previous studies will also originate from different times of the year when these studies were made.

All of our samples were collected on land, and regions that showed up as possible sources for highly ice active INP in Fig. 6 were on or close to land. While, as said above (Sec. 3.2.2) other regions can not be excluded as sources, it will be interesting to see in the future if $N_{\mathrm{INP}}$ detected further away from terrestrial sources will consistently be lower than those obtained on land.

Concerning an influence of INP emitted from ground on higher altitudes, Herenz et al. (2018) recently compared aerosol particle number size distributions (PNSD) measured on ground and during overflights at different heights in May 2014 in Tuktoyaktuk (Northwest Territories of Canada on the Arctic Ocean). PNSD measured on ground and in heights up 1200 m generally agreedwere similar, while PNSD obtained at larger altitudes were clearly different. Therefore, the atmospheric aerosol, including INP, can be similar in heights up to levels where cloud formation is observed. Hence, INP detected by ground-based measurements may well be able to influence ice formation in clouds, at least during times when the cloud layers are coupled to the surface.

Compared to previous literature data introduced in this study, the new long term data introduced in the here presented here study extends the range of Arctic $N_{\mathrm{INP}}$ towards higher $T$. It also clearly shows that Arctic INP concentrations vary throughout the year, with a regular presence of INP that are ice active at $T$ well above -10°C throughout the summer months at all four terrestrial measurement stations.

## 5  Discussion

The annual cycle observed for $N_{\mathrm{INP}}$ in this study is not in tune with what is known for particle number concentrations and size distributions occurring across the Arctic (Tunved et al., 2013; Nguyen et al., 2016; Freud et al., 2017). This is not too surprising: Recent studies, including Arctic locations, found that a large fraction of the number of INP are super-micron in size (Mason et al., 2016; Si et al., 2018; Creamean et al., 2018), while the majority of the number of particles are in the sub-micron size range.

Concerning the annual Arctic aerosol cycle, there is a maximum in particle number concentrations in early spring, before precipitation sets in, caused by accumulating anthropogenic pollution known as Arctic haze (Shaw, 1995). $N_{\mathrm{INP}}$ are low during that time of the year, which might indicate that anthropogenic pollution does not contribute to atmospheric INP, at least in the $T$ range examined in this study. This is in line with the observation that Arctic haze particles are not efficient INP (Borys, 1989), and also with a recent study showing that anthropogenic pollution did not contribute to INP in very polluted air in Beijing (Chen et al., 2018). It is also in agreement with observations by Hartmann et al. (2019), based on ice cores from Svalbard and Greenland, who found that $N_{\mathrm{INP}}$ in the Arctic did not increase over the past 500 years (from roughly 1480 to 1990), while tracers for anthropogenic pollution did increase markedly.

The formation of Arctic haze is related to the fact that during winter months, air masses from mid latitudes can transport aerosol particles into the Arctic (Heidam et al., 1999; Stohl, 2006). In contrast, in the summer months the Arctic lower atmosphere is effectively isolated and transport of atmospheric aerosol particles into the Arctic is low (Heidam et al., 1999; Stohl, 2006). Hence, the here observed increase in $N_{\mathrm{INP}}$ in late spring and summer has to originate from Arctic sources, including local ones situated close to the measurement site.

Once Arctic haze is removed by precipitation in spring (Browse et al., 2012), Arctic particles are mostly newly formed particles in the size range up to 100 nm (Lange et al., 2018), originating from gaseous precursors (Engvall et al., 2008; Leaitch et al., 2013; Croft et al., 2016; Wentworth et al., 2016; Dall'Osto et al., 2018). Number concentrations of these particles roughly peak in July and August (Freud et al., 2017). New particle formation ceases in late summer or early fall, and until Arctic haze

sets in again later in winter, the total particle number concentration is at its lowest, well below $100 \ \mathrm{cm}^{-3}$. But newly formed particles do not contribute to INP unless subject to particular conditions at cirrus level (Kanji et al., 2017). Nor does their annual cycle follow what we observed for $N_{\mathrm{INP}}$, as concentrations for newly formed particles cease much earlier in the year than observed for $N_{\mathrm{INP}}$.

Arctic concentrations of mineral dust particles have been measured and modeled (Fan, 2013; Zwaaftink et al., 2016) and

were described to largely depend on long-range transport, with largest concentrations occurring in spring. However, dust from local sources may also contribute (Zwaaftink et al., 2016). Mineral dust particles themselves likely only become ice active at $T$ at the lower end of or below those examined in the present study, but they may be carriers of biogenic ice active macromolecules (Conen et al., 2011; Tobo et al., 2014; O'Sullivan et al., 2014; Augustin-Bauditz et al., 2016; O'Sullivan et al., 2016; Hill et al., 2016). Continental sources of fungal spores were found to contribute to the organic aerosol observed in the Arctic in summer

(Fu et al., 2013), and it is known that spores of some fungal species are ice active at comparably high $T$ (Pummer et al., 2015). The same also holds for lichen (Moffett et al., 2015) and bacteria (Hartmann et al., 2013). Sources and abundance of bacteria were examined for an area in southwest Greenland (Santl-Temkiv et al., 2018), and roughly half of all airborne bacterial cells were described to originate from local terrestrial environments as e.g., surface soils, while the other half was said to have been long-range transported, originating from marine, glaciated, and terrestrial surfaces. In summary, microorganisms originating

from continental sources in general could contribute to the highly ice active INP observed in this study. This is in line with the observation presented in Fig. 6, where particularly for VRS and Ny Ålesund, highly ice active INP were observed for air masses that traveled over open land.

Particle generation at open leads in the Arctic has been observed (Held et al., 2011), related to sea spray from bubble production mechanisms that exist independently of wind (Norris et al., 2011). Leck and Bigg (2005) found particles in the

Arctic aerosol which could be attributed to algal exopolymer secretions and suggest they became airborne via bubble bursting. This agrees with observations by Orellana et al. (2011) and Fu et al. (2015), showing that Arctic marine microgels from algae, present both in surface water and SML, contribute to Arctic atmospheric particles. Such microgels were assumed to be related to INP observed in marine SML (Wilson et al., 2015). It was recently described that ice activity in Arctic seawater was negatively correlated with salinity (Irish et al., 2017), possibly indicating that these aquatic INPs were associated with

melting sea ice releasing ice active biogenic material into the ocean. Marginal ice zones are known to be of importance for phytoplankton blooms due to melting sea ice releasing iron (Wang et al., 2014). Overall, although the above mentioned particle production mechanism in Arctic open leads was found to be only of minor importance for the overall atmospheric particle number concentration (Held et al., 2011), this, together with wind driven sea spray production, could be a source for at least some of the highly ice active INP observed in our measurements. This likely holds for Alert, but it may also be the case in the

example of Utqiaġvik, where in April and May very little contributions of open land or open sea, as defined by the IMS maps,

was found, while highly ice active INP were present. These could potentially originate from open leads or polynyas. It should be added that Creamean et al. (2018) found a similarly large increase in highly ice active INP during May at Oliktok Point in northern Alaska, only roughly 300 km east of Utqiaġvik. This increase was related to INP from tundra surfaces and open water, particularly the marginal ice zone. Polynyas 700 km away from the measurement site were not found to contribute, likely due to settling of particles during the long traveling times in the slow moving air masses that were observed.

In summary, while anthropogenic pollution and new particle formation do not explain highly ice active INP observed in this study, these INP can originate both from terrestrial or marine sources in the Arctic. These sources are strong in summer and weak or absent in winter, depending on the conditions on the ground. Should terrestrial sources be found to contribute stronger than marine ones, $N_{\mathrm{INP}}$ further away from land will be lower than those observed here.

Concerning Arctic mixed-phase clouds at temperatures above -20°C, recently Norgren et al. (2018) observed a depression of cloud ice in these clouds when they are polluted, as observed during Arctic haze. Under these conditions, they have a lower amount of cloud ice mass for a given amount of condensed liquid mass. Based on the results of the here presented study, this can be explained by the lower concentrations of INP during times of Arctic haze - a reasoning that sheds light on the importance of the here reported findings which could not yet have been discussed in Norgren et al. (2018). With respect to recent modeling results, Solomon et al. (2018) did large-eddy simulations (LES) of Arctic mixed-phase stratocumulus clouds and find indications that changes in INP in the Arctic aerosol dominate over changes in cloud condensation nuclei (CCN). This shows the potential importance of determining INP and their changes due to climate change in the Arctic. However, Taylor et al. (2018) examined the Arctic annual cycle in cloud amounts from 24 different models and found significant disagreements. They conclude that the parameterization of the ice microphysics in the models contribute to the observed differences. Overall, the field of Arctic mixed-phase clouds and related ice nucleation currently is one of intense research, and the herein presented results contribute to our understanding and lay ground for improved descriptions of INP in modeling in the future.

## 6  Summary and Conclusions

This study shows a yearly cycle of $N_{\mathrm{INP}}$ at four different Arctic sites, an observation that has not been seen such clearly in previous studies. Maximum values were observed roughly from late spring until well into fall, minimum values were observed during winter and early spring. INP spectra for the most ice active INP are close to some observed at continental locations outside of the Arctic (Petters and Wright, 2015). Potential souce regions for INP that are ice active at the higher examined $T$ were determined based on combining air mass back-trajectories with IMS satellite maps. Such source regions were found in the Arctic on open land as well as on open water, particularly in the Baffin Bay, along the southern part of the west coast of Greenland and on Svalbard and the adjacent sea. Contributions from these regions could explain the increase in highly ice active INP observed in spring in Alert, Ny Ålesund and VRS. However, a possible source region for the highly ice active INP observed in Utqiaġvik could not be identified. Here, highly ice active INP appeared already earlier in the year than at the other stations. Open leads and polynyas present in the Arctic sea ice are not identified as open water in IMS maps and may have contributed to these INP.

Regions that were not identified as source regions for highly ice active INP in this study may still contribute such INP. It should still be mentioned that source regions determined here were mostly on or close to land. We showed that there is a large scatter in literature values observed for $N_{\mathrm{INP}}$ in the Arctic. As we find a yearly cycle in Arctic $N_{\mathrm{INP}}$, this large scatter may partially originate in different times of the year when samples were taken. It may also depend on the proximity to land where the sampling was done, if terrestrial sources and sources in close proximity to land should show to be dominant sourced for highly ice active INP in the future.

Independent of their origin, these observed INP might be transported certain distances and might have more than only local influences. This can become important, as biogenic emissions in general can be expected to increase in the Arctic in coming years due to Arctic amplification, which amplifies marine primary production (Arrigo et al., 2008; Ardyna et al., 2014), and changes Arctic microbial communities (Deslippe et al., 2012). Feedback mechanisms involving ice formation and with this radiative properties and lifetimes of Arctic stratiform clouds may exist, related to biogenic INP. Therefore, more thorough studies concerning Arctic INP are needed. Determining the current status of INP in the Arctic and future changes that are to be expected might help to understand aerosol cloud interactions in the Arctic and their significance for the observed strong Arctic warming.

*Data availability.* In case of acceptance, data will be made available on Pangaea.

*Author contributions.* H.W. and F.S. designed research; R.J.S., T.E.B., R.T., S.B., R.B., H.S., L.H., W.Z., and H.H. did field campaign management and/or filter sampling; T.E.B., C.E.M., and W.Z. did chemical analysis; M.H. calibrated and J.L., M.L., O.L. operated the ice nucleation measurement device; P.H. calculated trajectories; A.H. did the work on the satellite data; H.W. analyzed the data and wrote the paper with contributions from L.H., H.H., R.J.S., C.E.M., R.T., F.S., M.H., P.H., M.L., R.B., H.S. and W.Z..

*Competing interests.* There are no competing interests.

*Disclaimer.* ...

*Acknowledgements.* We gratefully acknowledge the funding by the Deutsche Forschungsgemeinschaft (DFG, German Research Foundation) – Projectnumber 268020496 – TRR 172, within the Transregional Collaborative Research Center "ArctiC Amplification: Climate Relevant Atmospheric and SurfaCe Processes, and Feedback Mechanisms (AC)[3]" for filter analysis and interpretations done in sub project B04. H.W. was partially funded by DFG within the Ice Nuclei research UnIT (INUIT, FOR 1525), WE 4722/1-2. We acknowledge Cecilia Shin, Darrell Ernst and Andrew Platt for logistic support, field work in Alert and assistance in analysis, and the Canadian Forces Station Alert for supporting sample collection. Logistic assistance of the Polar Support Unit of the CNR (Italian National Research Council) Department of Earth and Environment (POLARNET) in coordinating the activities based at the Dirigibile Italia Arctic station at Ny Ålesund is acknowledged, specifically Dr. Mauro Mazzola and Dr. Angelo Viola. Financial and technical support for samples from Utqiaġvik was provided by the United States Department of Energy (Atmospheric Radiation Measurement Field Campaign no. 2010-05876), NOAA (award no. NA14OAR4310150) and the C. Gus Glasscock, Jr. Endowed Fund for Excellence in Environmental Sciences. We also thank Walter Brower and Jimmy Ivanoff of the Ukpeagvik Inupiat Corporation for sample collection and field assistance and Fred Helsel, Dan Lucero, and Jeffrey Zirzow and the Sandia National Laboratory for site access and preparation. The work at Villum Research Station at Station Nord was financially supported by the Danish Environmental Protection Agency and Energy Agency via the MIKA/DANCEA funds for Environmental Support to the Arctic Region, which is part of the Danish contribution to the Arctic Monitoring and Assessment Programme (AMAP) and to the Danish research project "Short-Lived Climate Forcers" (SLCF). The Villum Foundation is acknowledged for funding the construction of Villum Research Station, Station Nord. The authors gratefully acknowledge the NOAA Air Resources Laboratory (ARL) for the provision of the HYSPLIT transport and dispersion model and/or READY website (http://www.ready.noaa.gov) used in this publication. And furthermore, we thank Richard Leaitch (ECCC) for helpful discussions during the writing process.

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

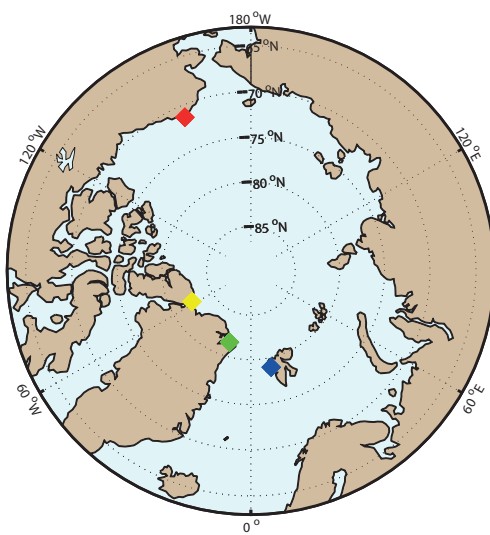

**Figure 1.** The location of the four stations from which filter samples were included herein: Utqiaġvik (red diamond), Alert (yellow diamond), VRS (green diamond) and Ny Ålesund (blue diamond).

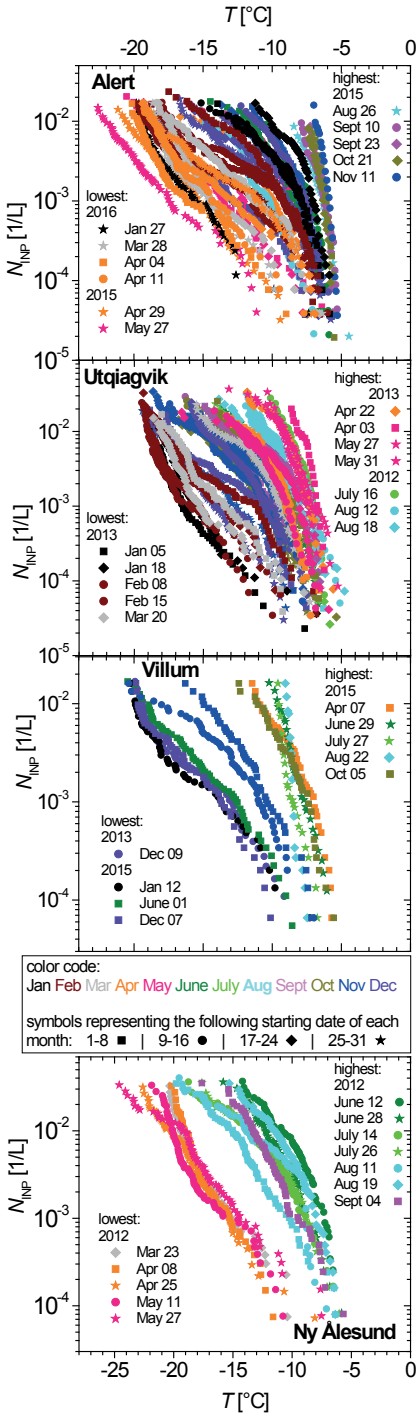

**Figure 2.** $N_{INP}$ for all samples. Symbol color distinguishes between data obtained for the different months, while symbol type indicates the day of the month when sampling started. Curves with particularly high and low $N_{INP}$ are explicitly listed in the legend. (The $T$ axis is the same for the upper three panels and different for the lower panel, see values for $T$ given on top and bottom of the figure, respectively.)

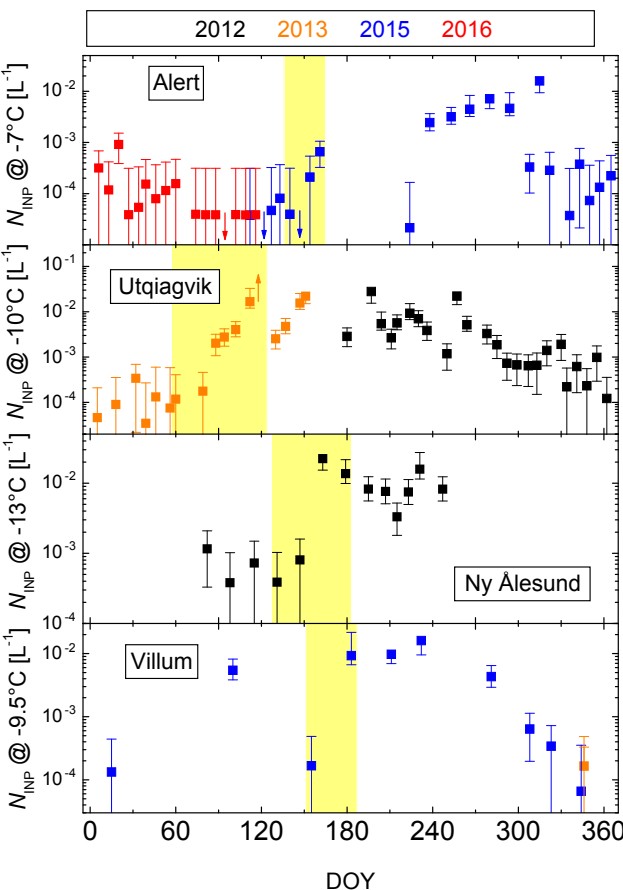

**Figure 3.** Time-series of $N_{\mathrm{INP}}$ at $T$ of -7°C, -10°C, -13°C and -9.5°C for Alert, Utqiaġvik, Ny Ålesund and VRS, respectively. Small arrows in the panels indicate times when values were either below (for Alert) or above (for Utqiaġvik) the detection limit. The yellow background shows for which samples an more in-depth analysis is presented in Sec. 3.2.2. Error bars show the 95% confidence interval.

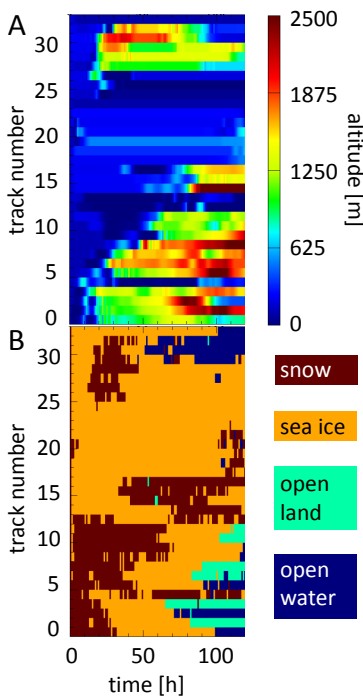

**Figure 4.** Exemplary results from the analysis of the IMS maps for the filter collected in Alert starting June 10, 2015. In both panels, each row (indicated as track number) represents one trajectory, going back in time for 5 days, starting from 0 (when sampling took place) and displaying 120 separate time steps. The colors indicate the altitude of the air mass for the different time steps in panel A and the respective nature of the ground at the location of the air mass in panel B.

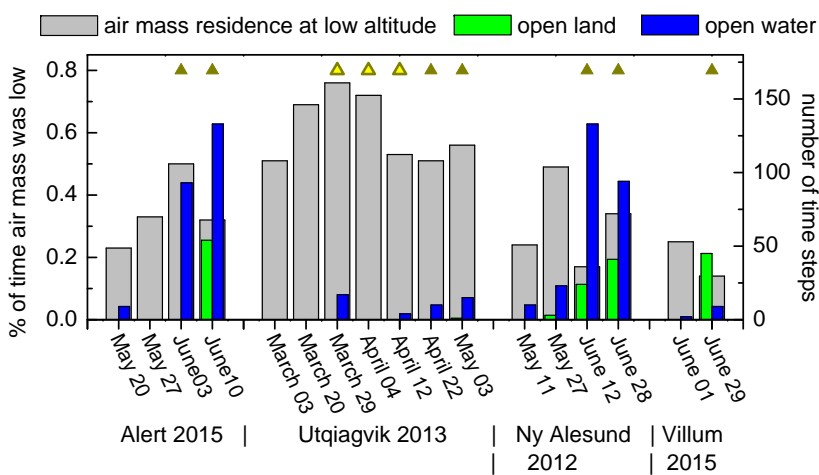

**Figure 5.** The number of time steps when air masses were at low altitudes and over open land or open water are shown in green and blue, respectively, for different filter samples. The analysis shown was done for altitudes up to 100 m for Alert, Ny Ålesund and VRS, and up to 500 m for Utqiaġvik. Gray bars in the background indicate the percentage of time the air masses collected on each filter were below that altitude. Triangles indicate samples for which highly ice active INP were detected, i.e., for which INP spectra were measured at high $T$ (single-colored triangles) and at medium $T$ (triangles with yellow interior). The respective INP spectra are shown in Fig 6.

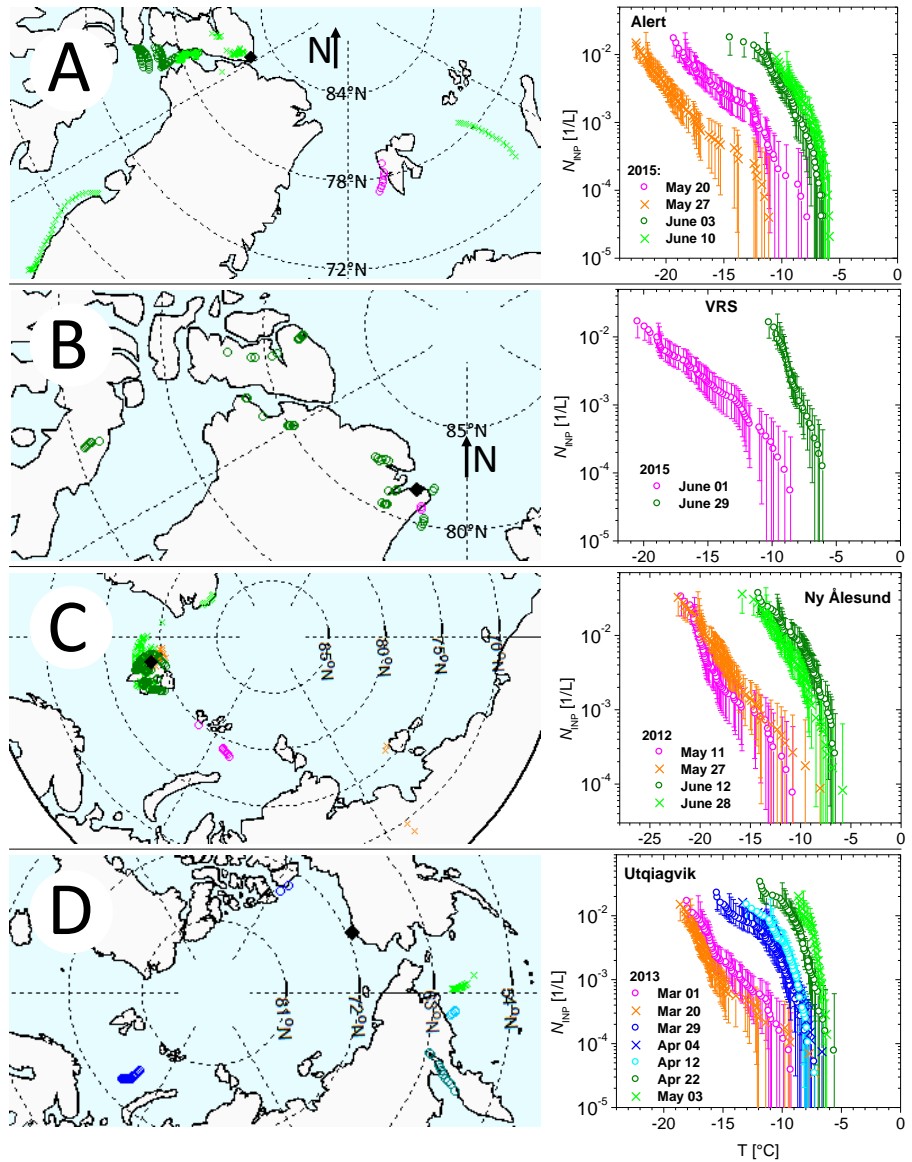

**Figure 6.** Rows A, B, C and D each show a map on the left side in which locations are indicated where air masses collected on different filters crossed over open land or open water while being at a low altitude (below 100 m for Alert, Ny Ålesund and VRS, below 500 m for Utqiaġvik). Black diamonds indicate the location of the measurement site. (Please note: the maps in the two lower rows are rotated by 90° compared to the those in the two upper rows.) The right panels show the INP spectra for the corresponding filters.

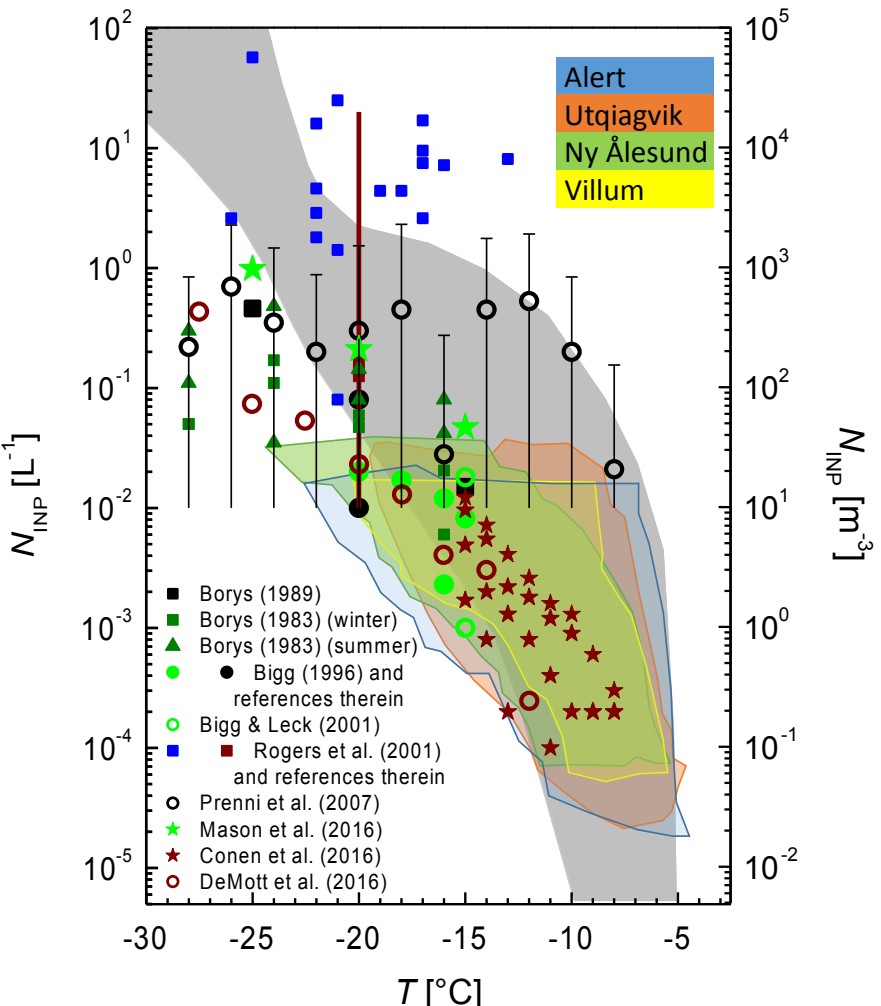

**Figure 7.** Comparison of $N_{INP}$ determined in this study for the Arctic with literature data by Petters and Wright (2015) (gray background) and Borys (1983, 1989); Bigg (1996); Bigg and Leck (2001); Rogers et al. (2001); Prenni et al. (2007); Mason et al. (2016); Conen et al. (2016); DeMott et al. (2016). Green and brown symbols represent data from groundsurface based measurements, black and blue ones from airborne measurements. For Rogers et al. (2001), brown data indicate data cited from literature, with the vertical bar indicating the extent of the reported values.

**Table 1.** R, $R^2$ and p values for linear correlations between $N_{\mathrm{INP}}$ (as shown in Fig. 2) and different bulk chemical properties.

| location | species | R | $R^2$ | p |
|---|---|---|---|---|
| Utqiaġvik | EC | -0.36 | 0.13 | 0.02 |
| | OC | -0.04 | <0.01 | 0.82 |
| | fluoride | 0.14 | 0.02 | 0.41 |
| | chloride | 0.13 | 0.02 | 0.43 |
| | nitrite | 0.22 | 0.05 | 0.19 |
| | bromide | -0.01 | <0.01 | 0.98 |
| | sulfate | -0.12 | 0.01 | 0.47 |
| | nitrate | 0.03 | <0.01 | 0.86 |
| Ny Ålesund | PM10 | -0.36 | 0.13 | 0.22 |
| | OC | -0.39 | 0.15 | 0.18 |
| | ammonium | -0.44 | 0.20 | 0.13 |
| | K | -0.57 | 0.32 | 0.04 |
| | Mg | -0.36 | 0.13 | 0.22 |
| | nitrite | 0.15 | 0.02 | 0.62 |
| | nitrate | -0.16 | 0.03 | 0.59 |
| | sulphate | -0.60 | 0.36 | 0.03 |
| | Na | -0.30 | 0.09 | 0.32 |
| | Ca | -0.50 | 0.25 | 0.08 |
| | Cl | -0.13 | 0.02 | 0.68 |
| | MSA | 0.18 | 0.03 | 0.55 |
| Alert | POC+CC | 0.59 | 0.35 | 0.01 |
| | OC | -0.12 | 0.01 | 0.62 |
| | EC | 0.05 | <0.01 | 0.84 |