# Peer review of "Annual variability of ice nucleating particle concentrations at different Arctic locations"

_Atmospheric Chemistry and Physics, 2018_

## Referee Comment (RC1) · Anonymous Referee #1 · 16 Jan 2019

General comments The manuscript presents longer term observations of INP concentrations at four Arctic stations. These are then discussed in the context of chemical compounds, back trajectories, and previously published data. The results are interesting, approach and methods are sound. The text is mostly clear but should be more concise. In particular the Introduction section has some lengths. I also wonder what the storage and transport histories of the samples from the field stations to the laboratory are meant to tell us. Atmospheric INP that could perish at room temperature would probably do so within a short time, perhaps already during a sampling period of several days. I can not imagine why it would help to freeze them again at a later stage. Possibly, many INP found in the atmosphere, even biogenic INP, can well be stored when dry at room temperature for many years (see Vasebi et al., 2019,

https://www.biogeosciences-discuss.net/bg-2018-496/#discussion).

The comparison of INP concentrations with chemical compounds in the same of air masses revealed no correlation. The discussion of that finding does not go much beyond current understanding, so it could be shortened substantially.

The back trajectory analysis to illuminate possible changes in source regions to explain observed increases in INP during spring or summer is original. I wonder how much the results of this analysis depend on the fact that a trajectory model was employed and not a particle dispersion model, in particular the results related to Utqiagvik.

The first part of section 3.3 (Determination of possible source regions; page 9, lines 5-33) should be moved into the Methods section.

Section 4 (comparison with literature) could be reduced in order to avoid repetition of what has been said already in the Introduction (or vice versa).

The Conclusion section would benefit from a couple of sentences relating back to the potential influence of INP on Arctic clouds stated in the Introduction. Do concentrations observed at any of the four stations fluctuate between concentrations that would have different effects on clouds?

Specific comments

Page 2, lines 13-15: "Not many measurements . . . . exist . . ." Either say that "measurements ...were done" or that "data . . . exist").

Site description: Please indicate the altitude of all four stations.

Page 5, line 6: ". . .the volume of air sampled per 1 mm piece of filter was roughly 270 or 540 L." Does this mean that you are not sure about the sample volume?

Page 8, lines 29-31: ". . .and therefore might be indicative of biogenic material. This might point towards a direction in which more detailed studies could be undertaken in the future." I wonder what substance is left of a thought after qualifying it with two

"might" and one "could".

Page 11, lines 29-30: "...N(INP) in the Arctic may, at times, be lower than at other locations." This statement probably applies to any location on Earth, except, if there is a location where N(INP) is always the global minimum N(INP).

Page 12, lines 30-31: "PNSD measured on ground and in heights up 1200 m generally agreed,..." Agreed on what? Or, do you mean "were similar"?

---

## Referee Comment (RC2) · Paul DeMott (Referee) · 30 Jan 2019

**General Comments**

This paper is a very nice to see effort to provide new ice nucleating particle concentration data for different Arctic locations, including long term coverage to establish seasonal and annual cycles. The use of samples collected for other compositional analyses provides additional utility. I will state upfront that I am involved in reporting of other measurements in this region, some of which are in review separately at this time, but have deemed that this does not color my review of this paper. I express some concerns about protocol and accounting for backgrounds, and I mention that I think that sections 3.2 and 3.3 can be combined as being about the same thing. In the discussions, I think

that some comments about the need for coarse mode aerosol measurements could be useful. And the authors do not need to emphasize local sources so much as they do I think. It seems unlikely that local is the only influence, although I understand that the sites used limit saying much more. It is the case that specific sources are difficult to discern. This is the case now in many studies. Nevertheless, this paper lays a nice groundwork for future research, providing additional impetus for new measurements over broad regions of the Arctic.

**Specific Comments**

**Abstract**

Page 1, Lines 9 and 10: Seems out of place for an abstract. Activity at this temperature range alone is not a certain indication of biogenic INPs, or at least should not be stated where that cannot be accompanied by discussion, in my opinion.

Page 1, Line 11: Reported generally, or where and when? It would help if context were given.

**Introduction**

Page 2, Lines 8 and 9: Meaning pure supercooled water clouds? And is this referring to a certain season or an annual basis?

Page 2, Line 11: "For primary ice formation in clouds, ice nucleation has to occur..." Somewhat awkward phrase and sentence. This is the definition of primary ice nucleation. That is, primary ice formation requires a heterogeneous ice nucleating particles. Perhaps rephrase?

Page 3, paragraph ending line 26: The study of Irish et al. (2019) seems relevant as well here (https://doi.org/10.5194/acp-19-1027-2019), especially with regard to land versus marine sources.

Page 3, lines 28 and 29: Sometimes spectra are linear, but you emphasize some data

that contradicts this (e.g., data more toward the upper bound of Petters and Wright, which is consistent with information in the book Microbiology of Aerosols, Chapter 3, Fig. 3.1.1). I understand where you are trying to go with this paragraph, but perhaps "typically"?

Page 3, lines 31-32: "But biogenic INP typically occur in low concentrations in the atmosphere." Regardless, the population may be entirely biogenic (i.e., dominant) at times over oceans on the basis of lab and field studies (e.g., McCluskey et al., 2018a,b).

Page 3, lines 34 and 35: Do mineral dusts always occur in higher concentrations than biogenics, even at -20 °C. It would seem to depend on the scenario of measurement, as one could imagine few or no dust INPs over some locations at times.

Page 4, line 3-4: There are "particularly high fractions of supercooled water observed in Arctic stratiform clouds" compared to where and when? Is there an expectation that seasons may matter, since open and ice-covered seas are present at different times?

Page 4, line 10: Should immersion mode be immersion freezing mode to be more explicit?

Page 5, line 1: Can you explain the meaning of blanks being collected? For example, does it mean they were removed temporarily from foil/wrap, placed in sampling shelter, returned to foil/wrap? That is, were they otherwise handled the same? And regarding storage after sampling at room temperature, were all samples stored that way and shipped after a certain time (e.g., weekly? annually?) Same for blanks?

Page 5, lines 5-8: The order of sentences describing procedures implies that filter punches were immersed in water prior to transit. Is this so? If not, rewrite this to be clear that a portion of the filter was sent to TROPOS first. What kind of water was used? Was the water tested for freezing both with and without a blank filter piece? If the punches were immersed and frozen before shipment, was blank water with a blank filter piece shipped?

Page 5, line 16: Same questions with regard to the Ny Ålesund filter samples. And another question might be if blank filters were also cut in the same manner with the same tools. These are just typical concerns regarding testing contamination due to processes used, and cutting with tools is a method not always used with filters.

Page 6, line 27: It is not clear from discussion here if the blank filter concentrations were used to adjust measurements or simply documented. Considering the statement on page 7, lines 16-17 that filters with 60 L volumes were not used due to background being too close to measurements, then it seems that many samples with 130 L volume (stated earlier) might also have sufficient background that a correction could be needed. This would seem to be important for establishing a clear lower bound to measurements.

**Results**

Page 8, lines 6-7: It seems not terribly surprising that over the limited T range assessed the curves do not intersect. There is no predicting what happens where measurements are not possible (e.g., lower T). The range measured is indeed the range where biogenic impacts generally and broadly occur, so I think this is expected, if that is the point.

Page 8, lines 11-12: While I can agree that the results suggest biogenic origins, it might be fair to point out that the studies referenced here conducted direct trials that more or less confirmed the nature of these sources as organic.

Page 8, Section 3.2: Although a statement is finally made about this later in the section (lines 32-33), one immediately wonders about the sufficiency of looking for correlations when the INPs represent an infinitesimal number fraction of all particles. It is a qualitative approach that many use, but it must only imply associations with certain air masses. I suggest to bring this point to the beginning of the section. This being clear, I wonder also about combining sections 3.2 and 3.3 under the general topic of associating INPs with air mass sources (e.g., source attribution of INP). The different

information is really targeting finding the same information. I wondered after reading both sections if joint information on composition and trajectories might carry more information than either one alone.

Page 9. Line 28: Overall, the analyses in this section end up being not conclusive, even though it is a reasonable idea to ask about the time air spends in the boundary layer over different surfaces, and so it is worthwhile. But is there a reason that 100 m was chosen, as opposed to some typical marine boundary layer depth over which the surface may be coupled? In this regard, it is not clear why 500 m was used for Utqiagvik, even though I looked for this information later. Just for contrast?

Page 9, line 32: precipitation "is assumed to lead"?

Page 10-11, paragraph: This paragraph wanders from a start talking about sea bird colonies, but I think the focus is that coastal regions seem particularly important as influencing INPs.

**Comparisons with literature**

Page 11, lines 29-30: Point taken, but I note that Petters and Wright (2013) contain data mostly from over NH continents. Delort and Amato, Eds. (2018; cf. Fig. 3.1.1) show ocean-based samples also in this lower region, as derived from DeMott et al. (2016), but for regions both inside and outside of the Arctic. Additional data in marine air-affected regions in Mcluskey et al. (2018b,c) would fall in these lower INP regions. Hence, it may be the case that many regions have yet to be effectively sampled, and not only that the Arctic is unique as a region of especially low INP.

Page 11, Fig. 7: Are the bars on the figure, horizontal and vertical, intended to represent uncertainties? For which set of data? It is a little unsatisfying not to see uncertainties represented somehow. It could be useful to show some. I can say that I am not entirely sure what the uncertainties are for Rogers et al. (2001), but given that the values are based on 10 s data, they could be quite large from a statistical sampling

standpoint. Uncertainties were characterized very well in the INP data set of Prenni et al., which are clearly measured at near the limit of detection of a CFDC. The confidence intervals on those data are reported in that paper, and they are quite large. In contrast, uncertainties on some of the immersion freezing data are relatively small. Is there any chance to represent some of these?

Page 12, last sentence: This statement may require a caveat that coupling and decoupling of the boundary layer from clouds, common at different times in the Arctic, will influence the conclusion.

**Discussion**

Page 13, lines 5-6: In speaking only about particle number concentrations, the possible need to consider size distributions is not discussed. And when size distributions are considered, it is apparent that most such data collected do not well capture the sizes of INPs that have recently been noted to be of most influence in the Arctic and coastal regions at the warmest activation temperatures (references already included). That is, most previous studies emphasize ultrafine and fine modes at sizes below 1 um, key for anthropogenic impacts, but ignoring the underappreciated coarse mode that may dominate INPs in this region. This seems an opportunity to mention this measurement need. You get to mentioning that ultrafine aerosols are not expected as INP sources, but these of course dominate total particle numbers and so should perhaps not be the focus of the start of discussion.

Page 14, line 19: Creamean et al. (2018) seems relevant to the suggestion about open leads.

Figure 7 caption: Measurements are referenced as ground based, but since some were over oceans, perhaps say "surface based".

**Supplemental**

Figure S1. This figure highlights significant overlap of the range of background freezing spectra and sample data in some cases. What is done for correcting the sample freezing spectra in times like April/May or other periods? Thinking here of Vali's recent discussion of these things in a recent AMTD paper (Vali, 2018).

SI, Page 9, line 10: Polen et al. (2018) seems relevant to the statement here.

SI, Page 9, lines 11-13: I understand and appreciate the feedback given in this small section overall, but the statements here could merit a lot of discussion and seem out of place in any case, which I say with bias, but also some justification. Why mention this disconnect between the usual sampling regimes of different INP measurement methods, if you are also saying they can be used in a complementary manner? It is clear that they have their use for ambient measurements in different parts of the INP temperature spectrum (DeMott et al., 2017), but also that aerosol pre-concentration can improve this overlap (Tobo et al., 2013 and others).

SI, Page 9, line 16: You may wish to be a little more explicit in the last parenthetic note. Heat sensitivity testing is possible with certain filter materials, and even for the same sample used for INP measurements. That is, a single filter of the right material can suit both standard freezing spectra tests and tests for heat lability if the rinsed suspension is divided. Readers may not know that quartz filters are less suitable for such tests. Your point is well taken of course that second filters could help with additional compositional measurements.

**Editorial Comments**

Page 1, lines 8 and 10: "months"

Page 1, line 9: "known"

Page 2, line 27: "among others observations, measurements of NINPÂăare needed"

Page 5, line 7: "from 4 to 13 days"

Page 5, lines 29-30: statement here somewhat repeats what is said earlier in the

paragraph

Page 7, line 11: "From this limitation, it also follows that…"

Page 7, line 23: "Sec. 4 and 5"

Page 8, line 22: "there were anti-correlations"

Page 13, line 1: "introduced in this study"

SI, page 9, line 4: "Price et al."

**References not already cited in paper**

DeMott, P. J., Hill, T. C. J., Petters, M. D., Bertram, A. K., Tobo, Y., Mason, R. H., Suski, K. J., McCluskey, C. S., Levin, E. J. T., Schill, G. P., Boose, Y., Rauker, A. M., Miller, A. J., Zaragoza, J., Rocci, K., Rothfuss, N. E., Taylor, H. P., Hader, J. D., Chou, C., Huffman, J. A., Pöschl, U., Prenni, A. J., and Kreidenweis, S. M.: Comparative measurements of ambient atmospheric concentrations of ice nucleating particles using multiple immersion freezing methods and a continuous flow diffusion chamber, Atmos. Chem. Phys., 17, 11227-11245, https://doi.org/10.5194/acp-17-11227-2017, 2017.

Irish, V. E., Hanna, S. J., Willis, M. D., China, S., Thomas, J. L., Wentzell, J. J. B., Cirisan, A., Si, M., Leaitch, W. R., Murphy, J. G., Abbatt, J. P. D., Laskin, A., Girard, E., and Bertram, A. K.: Ice nucleating particles in the marine boundary layer in the Canadian Arctic during summer 2014, Atmos. Chem. Phys., 19, 1027-1039, https://doi.org/10.5194/acp-19-1027-2019, 2019.

McCluskey, C. S.,ÂăHill, T. C. J., Sultana, C. M., Laskina, O., Trueblood, J., Santander; M. V., Beall, C. M., Michaud, J. M., Kreidenweis, S. M., Prather, K. A., Grassian, V. H., and DeMott, P. J., A mesocosm double feature: Insights into the chemical make-up of marine ice nucleating particles, J. Atmos. Sci.,Âă75, 2405-2423, doi: 10.1175/JAS-D-17-0155.1, 2018a.

McCluskey, C. S., Hill, T. C. J., Humphries, R. S., Rauker, A. M., Moreau, S., Strutton, P. G., Chambers, S. D., Williams, A. G., McRobert , I., Ward, J., Keywood, M. D., Harnwell, J., Ponsonby, W., Loh , Z. M., Krummel, P. B., Protat , A., Kreidenweis, S. M. and DeMott, P. J., Observations of ice nucleating particles over Southern Ocean waters. Geophysical Research Letters, 45, 11,989–11,997. https://doi.org/10.1029/2018GL079981, 2018b.

McCluskey, C. S., Ovadnevaite, J., Rinaldi, M., Atkinson, J., Belosi, F., Ceburnis, D., Marullo, S., Hill, T. C. J. , Lohmann, U., Kanji, Z. A., O'Dowd, C., Kreidenweis, S. M., and DeMott, P. J., Marine and Terrestrial Organic Ice Nucleating Particles in Pristine Marine to Continentally-Influenced Northeast Atlantic Air Masses, J. Geophys. Res. – Atmos., 123, 6196–6212. https://doi.org/10.1029/2017JD028033, 2018c.

Polen, M., Brubaker, T., Somers, J., and Sullivan, R. C.: Cleaning up our water: reducing interferences from nonhomogeneous freezing of "pure" water in droplet freezing assays of ice-nucleating particles, Atmos. Meas. Tech., 11, 5315–5334, https://doi.org/10.5194/amt-11-5315-2018, 2018.

Tobo, Y., Prenni, A.J., DeMott, P.J., Huffman, J.A., McCluskey, C.S., Tian, G., Pöhlker, C., Pöschl, U., Kreidenweis, S.M., 2013: Biological aerosol particles as a key determinant of ice nuclei populations in a forest ecosystem. J. Geophys. Res. -Atmos., 118, 10100-10110, doi:10.1002/jgrd.50801.

Vali, G.: Revisiting the differential freezing nucleus spectra derived from drop freezing experiments; methods of calculation, applications and confidence limits, Atmos. Meas. Tech. Discuss., https://doi.org/10.5194/amt-2018-309, in review, 2018.

---

## Author Comment (AC2) · 26 Mar 2019

Please find our responses to your review and the revised version of the manuscript inthe attachment.

Please also note the supplement to this comment:
https://www.atmos-chem-phys-discuss.net/acp-2018-1274/acp-2018-1274-AC2-supplement.pdf

---

## Author Response (AR1)

Dear Reviewer!

We thank you for doing this review and for your suggestions that helped to improve our manuscript. Below, please find your original comments in blue and our responses in black. As your suggestions often concern larger parts of the text that were mostly moved and only changed a little, we typically did not paste the new text into our answers below but instead point to where they can be found in the new version of the manuscript. This new version of the manuscript is attached at the end of this answer, with text that is going to be deleted printed in yellow and new text in blue, so that you can easily find the changes we made.

General comments

The manuscript presents longer term observations of INP concentrations at four Arctic stations. These are then discussed in the context of chemical compounds, back trajectories, and previously published data. The results are interesting, approach and methods are sound. The text is mostly clear but should be more concise. In particular the Introduction section has some lengths.

We thank you for the positive remarks. Following your suggestions below, some reorganization of the text was done. We agree that the Introduction section is detailed, but did not want to miss aspects related to the topic. The second reviewer (Paul DeMott) suggested, on a few occasions throughout the text, to even add citations or some more detail. We still feel that the text is organized in a better way, now, and hope you can agree with the changes we made.

I also wonder what the storage and transport histories of the samples from the field stations to the laboratory are meant to tell us. Atmospheric INP that could perish at room temperature would probably do so within a short time, perhaps already during a sampling period of several days. I can not imagine why it would help to freeze them again at a later stage. Possibly, many INP found in the atmosphere, even biogenic INP, can well be stored when dry at room temperature for many years (see Vasebi et al., 2019, https://www.biogeosciences-discuss.net/bg-2018-496/#discussion).

It is correct that the sample examined in the here cited discussion paper retained most of its ice nucleation ability (we have observed this too, in our lab, with a sample from Russel Schnell), but other biogenic INP are not so stable. There is e.g., a discussion around *Pseudomonas syringae*, which even when frozen has sometimes (but not always) been observed to lose its ice activity (Polen et al., 2016) - and from our own experience we know that it loses its ice activity quickly (in a matter of days to weeks) when it is kept in a refrigerator (> 0°C) or unfrozen. To the best of our knowledge, the full scale of changes that occur to ice activity of atmospheric samples during storage is not known in the community right now, but will certainly (have to) be addressed in the future. Therefore, freezing samples and reporting exactly what was done seems needed.

But as the transport of samples to TROPOS was similar in all cases, this part was summarized and is now only described once (in the first paragraph of Sec. 2 (now called "Methods") on the top of page 5), instead of repeating the almost similar text four times. We also added the reason given above for taking the precautions concerning cooling the samples.

The comparison of INP concentrations with chemical compounds in the same of air masses revealed no correlation. The discussion of that finding does not go much beyond current understanding, so it could be shortened substantially.

Following one of the second reviewer's remarks, we changed the order in this section and first mentioned the fact that generally no correlation is to be expected. Some correlations between INP and mineral dust particles in general were reported for Arctic samples in literature, and these are now mentioned, too.

The description of our results was shortened so that it reports in only one sentence that there is mostly no correlation. The remaining passage describes the results for POC+CC (pyrolyzed organic carbon and carbonate carbon) which we feel is justified, as this at least gives some indication of the nature of the INP.

For changes, see Sec. 3.2.1, starting on page 10, line 11.

The back trajectory analysis to illuminate possible changes in source regions to explain observed increases in INP during spring or summer is original. I wonder how much the results of this analysis depend on the fact that a trajectory model was employed and not a particle dispersion model, in particular the results related to Utqiagvik.

We used both, trajectories (combined with a potential source contribution function) and a particle dispersion model for tracking of sources of particles and cloud condensation nuclei for a study about data from the Antarctic recently (Herenz et al., 2019) and obtained similar results from both methods. Interestingly, back then, during the review process, it was rather deemed unnecessary by a reviewer to use both approaches. We cannot be certain that the same results would also be found for this new study discussed here for a dispersion model, but adding this second method here currently is beyond the scope of what can be done in the framework of this review.

The first part of section 3.3 (Determination of possible source regions; page 9, lines 5-33) should be moved into the Methods section.

We followed your recommendation, moving the first three paragraphs of Section 3.3 to the end of Sec. 2, which is now called "Methods" (instead of "Measurements"). Required slight changes were made in the text. For that please refer to the new version of the manuscript, Sec. 2.7 on page 8, line 20.

Section 4 (comparison with literature) could be reduced in order to avoid repetition of what has been said already in the Introduction (or vice versa).

There is, admittedly, some redundancy in the Introduction and Sec. 4. But while in the Introduction the literature was shortly introduced with respect to its content and scientific

outcome, in Sec. 4 it is mostly only discussed in terms of the INP concentrations or in connection to possible sources of INP. This section is one of the location where we were asked to include some more detail by the second reviewer. Therefore, for the time being, no deletions of text were done.

The Conclusion section would benefit from a couple of sentences relating back to the potential influence of INP on Arctic clouds stated in the Introduction. Do concentrations observed at any of the four stations fluctuate between concentrations that would have different effects on clouds?

We added the following paragraph at the end of the discussion section:

"Concerning Arctic mixed-phase clouds at temperatures above -20°C, recently Norgren et al. (2018) observed a depression of cloud ice in these clouds when they are polluted, as observed during Arctic haze. Under these conditions, they have a lower amount of cloud ice mass for a given amount of condensed liquid mass. Based on the results of the here presented study, this can be explained by the lower concentrations of INP during times of Arctic haze - a reasoning that sheds light on the importance of the here reported findings which could not yet have been discussed in Norgren et al. (2018). With respect to recent modeling results, Solomon et al. (2018) did large-eddy simulations (LES) of Arctic mixed-phase stratocumulus clouds and find indications that changes in INP in the Arctic aerosol dominate over changes in cloud condensation nuclei (CCN). This shows the potential importance of determining INP and their changes due to climate change in the Arctic. However, Taylor et al. (2018) examined the Arctic annual cycle in cloud amounts from 24 different models and found significant disagreements. They conclude that the parameterization of the ice microphysics in the models contribute to the observed differences. Overall, the field of Arctic mixed-phase clouds and related ice nucleation currently is one of intense research, and the herein presented results contribute to our understanding and lay ground for improved descriptions of INP in modeling in the future."

Based on what is described in this new paragraph, and particularly the results by Taylor et al. (2018), not enough is known, yet, to allow for more details on e.g., the concentrations of INP that are needed to influence these Arctic mixed-phase clouds.

Specific comments

Page 2, lines 13-15: "Not many measurements . . .. exist . . ." Either say that "measurements …were done" or that "data . . . exist").

Done.

Site description: Please indicate the altitude of all four stations.

Done.

Page 5, line 6: "...the volume of air sampled per 1 mm piece of filter was roughly 270 or 540 L." Does this mean that you are not sure about the sample volume?

We changed the formulation to: "… the volume of air sampled per 1 mm piece of filter differed for the different samples and varied from roughly 270 to 540."

Page 8, lines 29-31: ". . .and therefore might be indicative of biogenic material. This might point towards a direction in which more detailed studies could be undertaken in the future." I wonder what substance is left of a thought after qualifying it with two "might" and one "could".

This passage deals with POC (pyrolyzed organic carbon) which is rarely measured. But you are correct that the formulation here was overly cautious, and we changed it to: "This points towards a direction in which more detailed studies should be undertaken in the future."

Page 11, lines 29-30: ". . .N(INP) in the Arctic may, at times, be lower than at other locations." This statement probably applies to any location on Earth, except, if there is a location where N(INP) is always the global minimum N(INP).

You are totally right! We changed it to: "lowest Arctic $N_{INP}$, as those we observed in the winter months, might be below the lowest values observed on continents in mid-latitudes."

Page 12, lines 30-31: "PNSD measured on ground and in heights up 1200 m generally agreed,. . ." Agreed on what? Or, do you mean "were similar"?

Oups, a non-native-speaker issue, "were similar" is what we meant. Replacement done.

Literature mentioned in our answers:

Herenz, P., H. Wex, A. Mangold, Q. Laffineur, I. V. Gorodetskaya, Z. L. Flemming, M. Panagi, and F. Stratmann (2019), CCN measurements at the Princess Elisabeth Antarctica Research Station during three austral summers, Atmos. Chem. Phys., 19, 275–294, doi:10.5194/acp-19-275-2019.

Polen, M., E. Lawlis, and R. C. Sullivan (2016), The unstable ice nucleation properties of Snomax (R) bacterial particles, J. Geophys. Res.-Atmos., 121(19), 11666-11678, doi:10.1002/2016jd025251.

Dear Paul!

Thanks a lot for doing this review, for your thorough reading of our manuscript and for your encouraging words. The points you raised were mindful and certainly helped improving this manuscript. Below, please find your original comments in blue and our responses in black. Concerning the literature cited in this answer to your review, we ask you to refer to the attached new version of the manuscript (with tracked changes, showing text that is going to be deleted printed in yellow and new text in blue). Similaly, the new version of the supplemental information is also attached at the end.

**General Comments**

This paper is a very nice to see effort to provide new ice nucleating particle concentration data for different Arctic locations, including long term coverage to establish seasonal and annual cycles. The use of samples collected for other compositional analyses provides additional utility. I will state upfront that I am involved in reporting of other measurements in this region, some of which are in review separately at this time, but have deemed that this does not color my review of this paper. I express some concerns about protocol and accounting for backgrounds, and I mention that I think that sections 3.2 and 3.3 can be combined as being about the same thing. In the discussions, I think that some comments about the need for coarse mode aerosol measurements could be useful. And the authors do not need to emphasize local sources so much as they do I think. It seems unlikely that local is the only influence, although I understand that the sites used limit saying much more. It is the case that specific sources are difficult to discern. This is the case now in many studies. Nevertheless, this paper lays a nice groundwork for future research, providing additional impetus for new measurements over broad regions of the Arctic.

In our answers to your comments below, you will find our answers and the related changes we made concerning the points you raise there. On the issue of local sources, which did not come up in your specific comments again, we wonder where you feel that we stress them too much. Should you still want us to remove some mentioning of that, we would ask you to be more specific on where this should be done.

**Specific Comments**

**Abstract**

Page 1, Lines 9 and 10: Seems out of place for an abstract. Activity at this temperature range alone is not a certain indication of biogenic INPs, or at least should not be stated where that cannot be accompanied by discussion, in my opinion.

The sentence might have been misleading as it was not meant to say that we showed that the highly ice active INP are certainly biogenic in origin. We changed it such that this should be clear now:

"Although the nature of these highly ice active INP could not be determined in this study, it often has been described in literature that ice activity observed at such high temperatures originates from the presence of ice active material of biogenic origin."

Page 1, Line 11: Reported generally, or where and when? It would help if context were given.

This was meant to refer to the results from the section "Comparison with literature". It was changed to:

"Spectra observed at the lowest temperatures, i.e., those derived for winter months, were on the lower end of respective values from literature on Arctic INP and INP from mid-latitude continental sites, to which a comparison is presented herein."

**Introduction**

Page 2, Lines 8 and 9: Meaning pure supercooled water clouds? And is this referring to a certain season or an annual basis?

This was a try of a short summary of studies on different basis. To insert the requested information, this had to be extended. It now is:

"In the range of temperatures (T) down to -20°C, fractions of supercooled liquid clouds were reported to be well above 50%, based on annual mean data for Europe and North America (both including the Arctic) from satellite remote sensing (Choi et al., 2010). For a multi-year analysis of all clouds, based on ground-based remote sensing at two western Arctic locations (Eureka and Utqiaġvik), clouds containing only liquid water occurred at least 20% of the time in all months with a maximum of 56% in September (Shupe, 2011). Also during two Arctic aircraft campaigns operating out of Inuvik, each in April and May of two different years, based on in-situ measurements, at least 60% of the clouds observed down to -18°C were characterized as mostly liquid (Costa et al., 2017)."

Page 2, Line 11: "For primary ice formation in clouds, ice nucleation has to occur. . ." Somewhat awkward phrase and sentence. This is the definition of primary ice nucleation. That is, primary ice formation requires a heterogeneous ice nucleating particles. Perhaps rephrase?

Done, it now is: "Ice nucleation forms primary ice in clouds … "

Page 3, paragraph ending line 26: The study of Irish et al. (2019) seems relevant as well here (https://doi.org/10.5194/acp-19-1027-2019), especially with regard to land versus marine sources.

We added: "Irish et al. (2019) derived $N_{INP}$ during a ship cruise in the Canadian Arctic marine boundary layer in summer. They suggest that mineral dust contributed more strongly to the observed INP than sea spray, with mineral dust particles likely originating in the Arctic (Hudson Bay, eastern Greenland, north-west continental Canada)."

Page 3, lines 28 and 29: Sometimes spectra are linear, but you emphasize some data that contradicts this (e.g., data more toward the upper bound of Petters and Wright, which is consistent with information in the book Microbiology of Aerosols, Chapter 3, Fig. 3.1.1). I understand where you are trying to go with this paragraph, but perhaps "typically"?

The sentence in question was amended with: ", although at higher $T$ steep increases may be observed, followed by a weaker increase or even a plateau region down to roughly -20°C (e.g., Petters & Wright, 2015; O'Sullivan et al., 2018; Creamean et al., 2019)."

Page 3, lines 31-32: "But biogenic INP typically occur in low concentrations in the atmosphere." Regardless, the population may be entirely biogenic (i.e., dominant) at times over oceans on the basis of lab and field studies (e.g., McCluskey et al., 2018a,b).

You are right, and we extended this part as follows:

"Biogenic INP typically occur in low concentrations in the atmosphere, but nevertheless, at remote marine locations as the Southern Ocean, where $N_{INP}$ is generally low, marine biogenic INP might make up a large fraction or even the entire INP population (Burrows et al., 2013; McCluskey et al., 2018). At less remote locations, the majority of atmospheric INP consists of mineral dust particles originating from deserts or soils."

Page 3, lines 34 and 35: Do mineral dusts always occur in higher concentrations than biogenics, even at -20 _C. It would seem to depend on the scenario of measurement, as one could imagine few or no dust INPs over some locations at times.

Indeed, there are scenarios imaginable where there may be more biogenic INP than mineral dust INP that are ice active at -20°C, close to sources for biogenic INP or far away from sources for mineral dust particles (e.g., remote marine locations, see your comment above). We added "with the above mentioned exception of remote marine locations, typically".

Page 4, line 3-4: There are "particularly high fractions of supercooled water observed in Arctic stratiform clouds" compared to where and when? Is there an expectation that seasons may matter, since open and ice-covered seas are present at different times?

The sentence is more specific now: "The existence of particularly high fractions of supercooled water observed in Arctic stratiform clouds, as e.g., observed in Costa et al. (2017) in the temperature range above -20°C in comparison to more convective clouds in midlatitudes and the tropics, could be expected to be linked to a lack of biogenic INP in the Arctic, due to sparse biological activity."

Page 4, line 10: Should immersion mode be immersion freezing mode to be more explicit?

Done (here and in the following sentence).

Page 5, line 1: Can you explain the meaning of blanks being collected? For example, does it mean they were removed temporarily from foil/wrap, placed in sampling shelter, returned to

foil/wrap? That is, were they otherwise handled the same? And regarding storage after sampling at room temperature, were all samples stored that way and shipped after a certain time (e.g., weekly? annually?) Same for blanks?

We added the following clarifying information: "These field blanks were treated similar to the other filters, i.e., inserted into the sampler, however, without an airflow through them. They were also stored similarly to the sampled filters at all times."

Page 5, lines 5-8: The order of sentences describing procedures implies that filter punches were immersed in water prior to transit. Is this so? If not, rewrite this to be clear that a portion of the filter was sent to TROPOS first. What kind of water was used? Was the water tested for freezing both with and without a blank filter piece? If the punches were immersed and frozen before shipment, was blank water with a blank filter piece shipped?

The shipment is now only described once at the beginning of Sec. 3, as it was similar for samples from all four stations. In all cases, the filters were shipped, and punching and immersion in water was done in Leipzig directly prior to the measurements. We added at / moved to the end of the first paragraph in Sec. 2:

"Transport from the four institutes where the samples had been kept to TROPOS was done in a insulated boxes, together with cooling elements. The shipment was organized such that transport was fast (one to three days) and that upon arrival at TROPOS the temperature in the boxes was still below 0°C. At TROPOS, samples were again stored at -1°C until the measurements were done. These measures during storage and transport are precautions, as for biogenic INP, storage at temperatures above 0° C or even storage in general has been described to reduce their ice activity (Wex et al., 2015; Polen et al., 2016, respectively). In this study, until mentioned otherwise, all samples from all stations (including also field blanks) were treated similar during all procedures. In the next sections, peculiarities of the separate four stations are described, followed by details on the measurements and its evaluation."

The section you refer to here has been reworded for clarity:

"From these filters, a circular piece with 47 mm in diameter was shipped to Leipzig for this study.

The total sampling area on the filters was 17.8 cm x 22.8 cm. For the measurements at TROPOS, described in detail in Sec. 2.5 below, circles with 1 mm diameter were punched out from the samples using sterile biopsy punches, and immersed in ultra-pure water separately. The volume of air sampled per 1 mm piece of filter was differed for the different samples and varied from roughly 270 to 540 L."

As now said in the text, additional information on topics such as the water is (and had been) given in Sec. 2.5. See also our comment to your next to next remark.

Page 5, line 16: Same questions with regard to the Ny Ålesund filter samples. And another question might be if blank filters were also cut in the same manner with the same tools. These are just typical concerns regarding testing contamination due to processes used, and cutting with tools is a method not always used with filters.

Your comment here relates to topics we clarified above and we hope that you feel that all you are referring to here is sufficiently clear now.

 It is not clear from discussion here if the blank filter concentrations were used to adjust measurements or simply documented. Considering the statement on page 7, lines 16-17 that filters with 60 L volumes were not used due to background being too close to measurements, then it seems that many samples with 130 L volume (stated earlier) might also have sufficient background that a correction could be needed. This would seem to be important for establishing a clear lower bound to measurements.

This comment, combined with your comment on the treatment of the background wrt. the Supplemental Information (SI) was tackled by inserting a more detailed analysis on the treatment of the background to the SI. As this resulted in almost one additional page of text and an extra figure, we ask you to review the SI directly (attached at the end, following the revised version of the manuscript).

At the location you mention here, we additionally inserted the following:

"A subtraction of the signals of the field blank from those of the measurements was not done. This is justified in a detailed discussed in the SI, and the interpretation of the results from the filters presented in this study is the same for both uncorrected and background-corrected samples."

**Results**

Page 8, lines 6-7: It seems not terribly surprising that over the limited T range assessed the curves do not intersect. There is no predicting what happens where measurements are not possible (e.g., lower T). The range measured is indeed the range where biogenic impacts generally and broadly occur, so I think this is expected, if that is the point.

This passage justifies that results can be discussed based on measurements done at a single temperature, instead of having to discuss the entire measured INP spectra. But indeed, it is correct that there is no predicting what happens where measurements are not possible, and we reworded the first sentence in this paragraph to:

"It is worth noticing that once a sample has a comparably high concentration at one $T$ this is generally observed at all $T$ at which measurements are available, and vice versa, i.e., curves do not intersect much (see Fig. 2). Therefore, the curves shown in Fig. 3 can be used to discuss observed trends for INP that are ice active at high $T$."

Page 8, lines 11-12: While I can agree that the results suggest biogenic origins, it might be fair to point out that the studies referenced here conducted direct trials that more or less confirmed the nature of these sources as organic.

The sentences were changed to:

"These highly ice active INP will be in the focus of this study in the next two sections (Sec. 3.2 and 3.3). Similarly highly ice active INP have been suggested to be biogenic in origin based on

tests such as heat treatment (Hill et al., 2016; O'Sullivan et al., 2018), tests which due to the limited available amount of filter material could not be done in the present study."

Page 8, Section 3.2: Although a statement is finally made about this later in the section (lines 32-33), one immediately wonders about the sufficiency of looking for correlations when the INPs represent an infinitesimal number fraction of all particles. It is a qualitative approach that many use, but it must only imply associations with certain air masses. I suggest to bring this point to the beginning of the section. This being clear, I wonder also about combining sections 3.2 and 3.3 under the general topic of associating INPs with air mass sources (e.g., source attribution of INP). The different information is really targeting finding the same information. I wondered after reading both sections if joint information on composition and trajectories might carry more information than either one alone.

The last paragraph from the former Section 3.2 was moved to the beginning of this section, as you suggested. The following was added:

"At a lower $T$ of -25°C, recently Si et al. (2019) reported that mineral dust tracers correlated with INP, which suggests that mineral dust was a major contributor to the INP population at that $T$. Si et al. (2018) found that for three coastal sites in Canada a model based on K-feldspar as the only INP calculated INP concentrations that fit measurements well at -25°C while at -15°C measurements were under-predicted, suggesting a missing source of INP that are active at higher $T$. In the following it will be examined if there is a correlation between the INP that are ice active at high $T$ detected in the present study with chemical composition."

Former sections 3.2 and 3.3. were combined in the sense that a new Sec. 3.2 was formed ("Sources of INP") with these former sections being subsections in it. A further combination of the two topics was not done, as the correlation with chemical composition (as could be expected) hardly gave any insights, making a combination irrelevant.

Page 9. Line 28: Overall, the analyses in this section end up being not conclusive, even though it is a reasonable idea to ask about the time air spends in the boundary layer over different surfaces, and so it is worthwhile. But is there a reason that 100 m was chosen, as opposed to some typical marine boundary layer depth over which the surface may be coupled? In this regard, it is not clear why 500 m was used for Utqiagvik, even though I looked for this information later. Just for contrast?

Our analysis focuses on finding locations where emissions from open ground or open water could have taken place, and for this, the marine boundary layer depths does not play a role, as we are not looking into real clouds. As said:" An altitude restriction was used as we were trying to geographically locate INP sources on the surface." (This was in the part that now was moved to the new Sec. 2.7.) During the analysis for this study, we had looked at a number of different altitudes from 100m to 1000m and got quite similar results, independent of the altitude. Choosing 100m seems reasonable, as this allows for some uncertainty in the back-trajectories.

The extension to a larger altitude (and trajectories that reach further back) for Utqiagvik was done, as for this site, no apparent influence was found, independent of the altitude or length of the trajectories that were used (up to 10 days). This was to show that the altitude restrictions

and the choice of 5 day back-trajectories did not influence the analysis for this site. A passage of the text was reformulated to explain that better (new text in bold, here):

"Fig. 5 shows the number of time steps when air masses were over open land or open water for the separate filter samples. Additionally, gray bars in the background indicate the percentage of time the air masses collected on one filter were below 100 m for Alert, VRS and Ny Ålesund. It can already be seen, that the presence of highly ice active INP on a filter is related to air masses that fulfill the above given criteria, i.e., that traveled over open land or open water at a low altitude. It also can be seen that this was not found for Utqiaġvik. **Initially, no open land and hardly any open water had been found for this site when 5-day back-trajectories were used, together with an altitude restriction of 100 m, i.e., air masses did not travel over open land or open water at altitudes below 100 m. To check if the length of the back-trajectory or the chosen maximum altitude influenced our results for Utqiaġvik, an analysis was also done using 10-day back-trajectories and 500 m** as the altitude limit, **presented in Fig. 5. This extension** simply only resulted in larger percentages of time for which the air masses were below this altitude limit. But still not a large number of time steps was found for which air masses traveled over open land or open water for Utqiaġvik. We will get back to this again below."

Done. Please note that this is in the part that now was moved to the new Sec. 2.7.

 This paragraph wanders from a start talking about sea bird colonies, but I think the focus is that coastal regions seem particularly important as influencing INPs.

The text in this section was rearranged and reformulated:

"The above analysis shows that coastal regions may be particularly important as source for highly ice active INP, including open waters close to coasts. Indeed, highly ice active biogenic INP were found in Arctic surface waters before (e.g., Wilson et al., 2015; Irish et al., 2017). For the highly ice active samples collected on Ny Ålesund on June 12 and June 28, 2012, the surrounding of the measurement station was completely snow free during the times when these samples were collected, whereas for all other cases there was at least partial or total snow cover around the stations. In other words, local terrestrial sources close to the measurement station may also contribute as sources for highly ice active INP, as already discussed in Creamean et al. (2018). Also Irish et al. (2019) describe Arctic land masses to be the source for observed Arctic INP (ice active at -15°C, -20°C and -25°C), and these INP were suggested to be mineral dust. On Svalbard, Tobo et al. (2019) found higher atmospheric $N_{INP}$ in July than in March, and they additionally described glacial outwash sediments in Svalbard to be highly ice active. This ice activity was assumed to be connected to small amounts of organic (likely biogenic) material. Based on these findings, Tobo et al. (2019) suggest the higher $N_{INP}$ in summer to be connected to organic (biogenic) components in glacially sourced dust. Some coastal regions in the Arctic, e.g., the west coast of Greenland together with the region around Baffin Bay and the Canadian Arctic Archipelago as well as the area around the Bering Strait and also Svalbard are known for their abundance of sea bird colonies (Croft et al., 2016). These regions partially coincide with regions highlighted as possible INP sources in Fig. 6. These

regions are known to emit ammonia, which plays a role in new particle formation in the Arctic (Croft et al., 2016). But clearly, newly formed particles are not expected to contribute to atmospheric INP at the temperatures examined in this study, and INP are likely simply also emitted from regions with high biological activity. In Sec. 5 we will discuss possible INP sources in more detail."

**Comparisons with literature**

Page 11, lines 29-30: Point taken, but I note that Petters and Wright (2013) contain data mostly from over NH continents. Delort and Amato, Eds. (2018; cf. Fig. 3.1.1) show ocean-based samples also in this lower region, as derived from DeMott et al. (2016), but for regions both inside and outside of the Arctic. Additional data in marine air-affected regions in Mcluskey et al. (2018b,c) would fall in these lower INP regions. Hence, it may be the case that many regions have yet to be effectively sampled, and not only that the Arctic is unique as a region of especially low INP.

This passage was revised and now is:

"At the lowest temperatures at which we detected INP spectra, $N_{INP}$ are lower than data from Petters and Wright (2015), i.e., lowest Arctic $N_{INP}$, as those we observed in the winter months, might be below the lowest values observed on continents in mid-latitudes. It is worthwhile adding that still lower concentrations were observed in marine remote locations in the Southern Ocean (McCluskey et al., 2018a) and for clean marine air in the North East Atlantic (McCluskey et al., 2018b)."

Page 11, Fig. 7: Are the bars on the figure, horizontal and vertical, intended to represent uncertainties? For which set of data? It is a little unsatisfying not to see uncertainties represented somehow. It could be useful to show some. I can say that I am not entirely sure what the uncertainties are for Rogers et al. (2001), but given that the values are based on 10 s data, they could be quite large from a statistical sampling standpoint. Uncertainties were characterized very well in the INP data set of Prenni et al., which are clearly measured at near the limit of detection of a CFDC. The confidence intervals on those data are reported in that paper, and they are quite large. In contrast, uncertainties on some of the immersion freezing data are relatively small. Is there any chance to represent some of these?

The blue vertical bar showed an average value and was removed as the separate data points from Rogers et al. (2001) are shown, too. The horizontal brown bar shows the range of literature data cited by Rogers et al. (2001), which is now explicitly mentioned in the figure caption. Error bars from Prenni et al. (2007) are now shown in Fig. 7 and the following text was added to the manuscript:

"For the data taken from (Prenni et al., 2007), the reported measurement uncertainty is shown, representative for typical uncertainties for the type of instrumentation used in Rogers et al. (2001) and Prenni et al. (2007). Error bars indicate one standard deviation at the higher end. The lower end is indicative of the detection limit, and for a substantial fraction of measurements no INP were detected in Prenni et al. (2007)."

As typical uncertainties for the immersion freezing instrumentation used in this study were shown in Fig. 3 and 6 already, they were not added again in Fig. 7, to not overcrowd the latter.

Page 12, last sentence: This statement may require a caveat that coupling and decoupling of the boundary layer from clouds, common at different times in the Arctic, will influence the conclusion.

The sentence was complemented and now is:

"Hence, INP detected by ground based measurements may well be able to influence ice formation in clouds, at least during times when the cloud layers are coupled to the surface."

**Discussion**

Page 13, lines 5-6: In speaking only about particle number concentrations, the possible need to consider size distributions is not discussed. And when size distributions are considered, it is apparent that most such data collected do not well capture the sizes of INPs that have recently been noted to be of most influence in the Arctic and coastal regions at the warmest activation temperatures (references already included). That is, most previous studies emphasize ultrafine and fine modes at sizes below 1 um, key for anthropogenic impacts, but ignoring the underappreciated coarse mode that may dominate INPs in this region. This seems an opportunity to mention this measurement need. You get to mentioning that ultrafine aerosols are not expected as INP sources, but these of course dominate total particle numbers and so should perhaps not be the focus of the start of discussion.

The opening paragraph of the Discussion section now cites the recent literature concerning this matter to stress the point you make here:

"The annual cycle observed for $N_{INP}$ in this study is not in tune with what is known for particle number concentrations and size distributions occurring across the Arctic (Tunved et al., 2013; Nguyen et al., 2016; Freud et al., 2017). This is not too surprising: Recent studies, including Arctic locations, found that a large fraction of the number of INP are super-micron in size (Mason et al., 2016; Si et al., 2018; Creamean et al., 2018), while the majority of the number of particles are in the sub-micron size range."

The paragraph dealing with new particle formation was shorted by one sentence.

Page 14, line 19: Creamean et al. (2018) seems relevant to the suggestion about open leads.

Done: "It should be added that Creamean et al. (2018) found a similarly large increase in highly ice active INP during May at Oliktok Point in northern Alaska, only roughly 300 km east of Utqiaġvik. This increase was related to INP from tundra surfaces and open water, particularly the marginal ice zone, while polynyas 700 km away from the measurement site were not found to contribute, likely due to settling of particles during the long traveling times in the slow moving air masses that were observed."

Figure 7 caption: Measurements are referenced as ground based, but since some were over oceans, perhaps say "surface based".

Done.

**Supplemental**

Figure S1. This figure highlights significant overlap of the range of background freezing spectra and sample data in some cases. What is done for correcting the sample freezing spectra in times like April/May or other periods? Thinking here of Vali's recent discussion of these things in a recent AMTD paper (Vali, 2018).

We went back to check this very carefully. This is treated in much more detail now, in the first section of the SI (adding ~ one page of text and a figure), and we refer you to that to check if you can agree with our treatment of the background and the reasoning behind it.

SI, Page 9, line 10: Polen et al. (2018) seems relevant to the statement here.

We incorporated this citation.

SI, Page 9, lines 11-13: I understand and appreciate the feedback given in this small section overall, but the statements here could merit a lot of discussion and seem out of place in any case, which I say with bias, but also some justification. Why mention this disconnect between the usual sampling regimes of different INP measurement methods, if you are also saying they can be used in a complementary manner? It is clear that they have their use for ambient measurements in different parts of the INP temperature spectrum (DeMott et al., 2017), but also that aerosol pre-concentration can improve this overlap (Tobo et al., 2013 and others).

The point under discussion here was intended to say exactly what you say in this remark, i.e., that in-situ devices and filter-offline techniques operate in different sampling regimes (based on INP concentrations), and that, in this sense, they are complementary. It is formulated now in a positive way, including the possibility of pre-concentration:

"However, they need comparably high $N_{INP}$ to overcome their detection limits and hence typically contribute values at lower $T$ where $N_{INP}$ is higher. Concentration of the aerosol prior to the in-situ sampling, done e.g., in Tobo et al. (2013) can help to increase that $T$ up to which measurements can be made, increasing the range of $T$ for which overlap between in-situ and off-line techniques can be obtained."

SI, Page 9, line 16: You may wish to be a little more explicit in the last parenthetic note. Heat sensitivity testing is possible with certain filter materials, and even for the same sample used for INP measurements. That is, a single filter of the right material can suit both standard freezing spectra tests and tests for heat lability if the rinsed suspension is divided. Readers may not know that quartz filters are less suitable for such tests. Your point is well taken of course that second filters could help with additional compositional measurements.

The text here was extended:

"It should be added that also the choice of filter material is important. Polycarbonate filters can be washed off and different analysis, including heat treatment, can be done on the suspensions. Teflon filters were found to not work well for washing off in Chen et al. (2018) and also could not be punched. When considering heat treatment, parts of sampled quartz fiber filters could be heated prior to punching filter pieces for the analysis, while suspensions cannot be made as INP are retained by the fibers (Conen et al., 2012)."

**Editorial Comments**

Please note: when these comments only required the addition of a letter or a word, these changes were not always highlighted in the new version of the manuscript.

Page 1, lines 8 and 10: "months" Done.

Page 1, line 9: "known" Done.

Page 2, line 27: "among others observations, measurements of NINP¡aare needed" Done.

Page 5, line 7: "from 4 to 13 days"
Sorry, but it is not clear what you mean. This line talks about the duration of the transport from the institutes which originally collected and stored the samples to TROPOS, and this never took longer than 3 days (even much shorter for the samples from Europe). (Special transport was arranged for these shipments.) 4 to 13 days were the sampling times at Utqiagvik.

Page 5, lines 29-30: statement here somewhat repeats what is said earlier in the paragraph
Information on the transport to TROPOS, which was repeated again separately four times, was summarized and moved to the end of the first paragraph of section 2.

Page 7, line 11: "From this limitation, it also follows that. . ." Done.

Page 7, line 23: "Sec. 4 and 5" Done.

Page 8, line 22: "there were anti-correlations" Done.

Page 13, line 1: "introduced in this study" Done.

SI, page 9, line 4: "Price et al." Done.

[revised manuscript text omitted]

by Wex et al.

**Correspondence:** Heike Wex (wex@tropos.de)

**1 Background measurements**

Following a recommendation given in Polen et al. (2018), Figs. S1 to S3 show frozen fractions ($f_{ice}$), i.e., the measured parameter, for field blanks together with some spectra of $f_{ice}$ for pure water and for filter samples that were sampled in the days directly before and after the field blank was taken. Field blanks were treated similar to filters onto which sampling was done, only were they not subjected to sampling air through them. They did, however, spend time in the sampler. The three different panels in Fig. S1 clearly show, that the background level is influenced by the atmospheric INP concentrations, as field blanks taken from April until October show a much larger signal than those collected during the other months. Therefore, and as the availability of field blanks varied between the different measurement stations (there were none for VRS, two for Ny Ålesund and Utqiaġvik each, and 9 from Alert), signals from the field blanks were not subtracted from the signals from the samples. However, in all cases and at all stations, the background was low enough to not have influenced the interpretation of the results from the filters presented in this study.

To show the influence of the background on the measurements in more detail, for five filter samples from Alert the background was subtracted. These five samples were selected such that the background had the highest possible impact. From the upper panel of Figs. S1, both samples were included (April 29 and May 07), relating to the same field blank which had been taken on May 07. From the middle panel, the higher one of the two field blanks was taken, together with the lower one of the two samples that it was collected closest to (October 21). Similarly, from the lowest panel, the highest field blank with the lowest of the two related samples (June 10) and the lowest sample (January 27) with the related field blank were included. The five spectra of $f_{ice}$ for the filter samples are shown together with those for the field blanks in the two left panels of Fig. S4. Subtraction of the background was done by converting $f_{ice}$ to concentrations of INP per volume of water/suspension (known as K(T) in the nomenclature by Vali, 2019), following Polen et al. (2018). K(T) from the field blanks was then subtracted from that of the filter samples, and the result was converted to background corrected atmospheric INP number concentrations. Ultimately this procedure can be summarized as:

$$N_{INP,corr} = (-ln(1 - f_{ice,s}) + ln(1 - f_{ice,b}))/V \tag{1}$$

The corrected atmospheric INP number concentration is $N_{INP,corr}$, the frozen fractions measured for the filter samples and the field blanks are $f_{ice,s}$ and $f_{ice,b}$, respectively, and $V$ is the volume of air sampled onto a 1 mm filter piece which was immersed in each examined droplet.

Uncorrected and corrected INP concentrations are shown in the two right panels of Fig. S4. $N_{INP,corr}$ was lower than the uncorrected value by less than 2% for two samples (June 03 and October 21) and between 5% and 20% for January 27 and for the samples from April 29 and May 07 below -14°C. Above -14°C, this difference was between 20% and 40% for May 07, which, compared to the range the overall signals span, is still small and within the uncertainty given in the main text. For the sample from April 29 above -14°C, there are clear differences between uncorrected and corrected values, going up to 80%, and at the highest temperatures there were two values for which the field blank showed higher $f_{ice}$ than the sample (see red circle in lower right panel). The respective field blank had been taken on May 07, i.e., at the end of the period when the sample from April 29 was collected. Also, the sample from April 29 was amongst the lowest measured (it could not be included in the time series shown in Fig. 3 in Sec. 3.1 in the main text). Indeed, the sample of April 29 was the only one in this whole study for which a background signal was above the respective measurement. As it was shown above, the signal from the field blanks varied along with the signals from the sampled filters. The sample from April 29 was collected at the end of the season with low INP concentrations, and possibly the field blank taken on May 07 does not represent the background that was present during the time when the sample from April 29 was collected. Therefore, for the case of April 29, rather the attribution of this field blank to the filter, rather than the data obtained from that filter itself, might be considered problematic.

With a background as variable as that observed here, subtracting a background may induce errors. A subtraction of the background was not done in this study, for the following reasons: 1) As discussed above, even for these examples with a comparably high influence from the background, the difference between corrected and uncorrected INP concentrations was generally low. 2) A clear attribution of field blanks to related samples would, in all cases, be needed, due to the variable background, but this seems not even possible for Alert, for which the largest amount of field blanks was available. 3) The interpretation of the results from the filters presented in this study is the same for both uncorrected and background-corrected samples.

Some filters sampled during spring 2017 in Alert and from March until September 2015 in Ny Ålesund also had been examined. However, these filters had shorter sampling times of only one day and less than 60 L of air had been sampled onto each single circular 1 mm filter piece. $f_{ice}$ determined for these filters were so close to the background determined from the blank filters, and therefore that these data were not used in this study. It should be added that for cases where a lower background can be achieved or where air with high concentrations of INP is sampled, 60 L of air sampled into each one of the examined droplets may suffice to get values that are well enough separated from the background.

[Figure]

**Figure S1.** $f_{ice}$ derived for measurements of pure water (blue open circles), blank filters (black circles with yellow filling) and filter samples sampled prior and after the blank filter was taken. The color code for the filter samples is the same as used in Fig. 1.

[Figure]

**Figure S2.** Similar to Fig. S1, but for Ny Ålesund.

[Figure]

**Figure S3.** Similar to Fig. S1, but for Utqiaġvik.

[Figure]

**Figure S4.** The two panels to the left show frozen fractions measured for five different filter samples (open circles), together with the respective field blanks (diamonds, filled in yellow). Symbol colors indicate the day when sampling of the filter sample was started. (Two separate panels were used to increase the visibility of the separate curves.) The two right panels show uncorrected and background-corrected INP concentrations (open circles and stars, respectively). The red circle in the lower right panel shows the only time observed during this study when a background signal was above the measured signal.

**2 Back-trajectories**

Figs S6 to S8 show the back-trajectories that were derived for the 17 selected filters examined in detail in Sec. 3.3. For Utqiaġvik, 5-day and 10-day back-trajectories are shown separately in Figs S7 and S8 and the information on the altitude of these back-trajectories is explicitly shown in Fig. S9.

[Figure]

**Figure S5.** 5-day back-trajectories for the filter samples from Ny Ålesund discussed in Sec. 3.3.

[Figure]

**Figure S6.** Similar to Fig. S5, but for Alert and VRS.

[Figure]

**Figure S7.** Similar to Fig. S6, but for Utqiaġvik.

[Figure]

**Figure S8.** Similar to Fig. S7, but showing 10-day back-trajectories.

[Figure]

**Figure S9.** Altitudes of the 10-day back-trajectories displayed in Fig. S8.

**3 Recommendations**

We felt it could help future research if we shared some recommendations, based on lessons we learned. These are the following:

- It could be advantageous to sample on filters that allow for washing off particles, as this enables to do dilution series. With this, obtained data can cover a broader $T$ range (e.g., polycarbonate membrane filters, Price et al., 2018), compared to that obtained in the present study.

- A higher time resolution used in the filter sampling will facilitate source apportionment. Still, care has to be taken to sample enough material to be above the detection limit, as for the present study samples on which less than 60 L of air had been sampled onto each single circular 1 mm filter piece (i.e., into each examined droplet) could not be used as they were too close to the background (see above, SI 1). Therefore shorter sampling times have to be counterbalanced by higher flow rates during sampling. Suppressing the filter background, if possible, would be of advantage, too. Recently, Polen et al. (2018) gave a number of recommendations related to use of and data from droplet freezing techniques as the one applied in this study, with a focus on working cleanly.

- Sampling with in-situ devices (Rogers et al., 2001; Prenni et al., 2007) can complement off-line filter analysis, measuring down to lower $T$. However, they need comparably high $N_{INP}$ to overcome their detection limits and hence typically do not obtaincontribute values at higherlower $T$ where $N_{INP}$ is higherlower. Concentration of the aerosol prior to the in-situ sampling, done e.g., in Tobo et al. (2013), can help to increase that $T$ up to which measurements can be made, increasing the range of $T$ for which overlap between in-situ and off-line techniques can be obtained.

- Parallel sampling of additional sufficient material to derive chemical composition enables more indepth testing of possible components present in INP. This might help to connect INP to their sources or to at least enable to corroborate the biogenic nature of those INP active at high $T$, using e.g., a test of the heat sensitivity of INP for the latter. It should be added that also the choice of filter material is important. Polycarbonate filters can be washed off and different analysis, including heat treatment, can be done on the suspensions. Teflon filters were found to not work well for washing off in Chen et al. (2018) and also could not be punched. When considering heat treatment, parts of sampled quartz fiber filters could be heated prior to punching filter pieces for the analysis, while suspensions can not be made as INP are retained by the fibers (Conen et al., 2012).